# Open Your Eyes:
# Vision Enhances Message Passing Neural Networks in Link Prediction

**Yanbin Wei** [1 2]  **Xuehao Wang** [1]  **Zhan Zhuang** [1 3]  **Yang Chen** [1]  **Shuhao Chen** [1]  **Yulong Zhang** [4]  **James Kwok** [2]  **Yu Zhang** [1]

## Abstract

Message-passing graph neural networks (MPNNs) and structural features (SFs) are cornerstones for the link prediction task. However, as a common and intuitive mode of understanding, the potential of visual perception has been overlooked in the MPNN community. For the first time, we equip MPNNs with vision structural awareness by proposing an effective framework called Graph Vision Network (GVN), along with a more efficient variant (E-GVN). Extensive empirical results demonstrate that with the proposed framework, GVN consistently benefits from the vision enhancement across seven link prediction datasets, including challenging large-scale graphs. Such improvements are compatible with existing state-of-the-art (SOTA) methods and GVNs achieve new SOTA results, thereby underscoring a promising novel direction for link prediction. The official code is available at https://github.com/WEIYanbin1999/EGVN.

## 1. Introduction

Link prediction, which predicts the presence of a connection between two nodes, is a core task in graph machine learning. It is crucial in numerous applications such as recommendation systems (He et al., 2020), drug interaction prediction (Yamanishi et al., 2008), and knowledge-based reasoning (Bordes et al., 2013; Wei et al., 2023).

Message-passing graph neural network (MPNN) (Kipf & Welling, 2017; Hamilton et al., 2017) is a prominent and powerful tool for link prediction. It generates node repre-

sentations and aggregates them into link representations to predict the existence of links. Besides, various structural features (SFs), which are characteristics derived from the graph topology to capture relationships and properties of nodes and edges, are integrated into MPNNs to enhance their link prediction capabilities (Zhang & Chen, 2018; You et al., 2021; Zhu et al., 2021; Yun et al., 2021; Chamberlain et al., 2023; Wang et al., 2024). This integration led to significant advancements, making the SF-enhanced MPNNs dominate the link prediction task.

Considering human perception of graph data, one of the most crucial means is intuitively interpreting of graph information through visual perception. While the structural awareness in MPNNs is derived from message-passing paths, and SFs are typically based on specific heuristic priors, both approaches have overlooked the potential of utilizing visual modalities to comprehend graph structures. As a fundamental capability of human perception, vision has matured into a thriving field supported by robust research tools, such as readily available visual encoders such as VGG16 (Simonyan & Zisserman, 2015), ResNet50 (He et al., 2016), and ViT (Dosovitskiy et al., 2021). These tools offer powerful capabilities for visual information awareness and extraction. Various research domains, including natural language processing, have successfully integrated visual modalities, leading to significant advancements in areas such as visual question-answering (Antol et al., 2015) and multimodal large language models (Zhang et al., 2024).

Given these developments, it is both timely and logical to consider incorporating vision into MPNNs. To be specific, we focus on integrating visual graph awareness into the MPNN architecture and exploring the roles that vision can play in enhancing link prediction. To this end, we propose three important but unexplored research questions:

- **RQ1**: *For the link prediction task, what is the effective approach to be aware of graph structures from vision?*

- **RQ2**: *Does incorporating vision awareness into graph structures enhance link prediction? If so, what are the underlying reasons?*

[1]Southern University of Science and Technology, Shenzhen, China [2]Hong Kong University of Science and Technology, Hong Kong, China [3]City University of Hong Kong, Hong Kong, China [4]Zhejiang University, Hangzhou, China. Correspondence to: Yu Zhang <yu.zhang.ust@gmail.com>.

*Proceedings of the 42nd International Conference on Machine Learning*, Vancouver, Canada. PMLR 267, 2025. Copyright 2025 by the author(s).

- **RQ3**: *If vision-based structural awareness proves beneficial, how can it be effectively fused into the MPNN architecture?*

To address the above research questions, we propose a novel framework called **G**raph **V**ision **N**etwork (**GVN**), as well as a more efficient variant **E-GVN**. The proposed models seamlessly incorporate visual awareness of graph structures into MPNNs for link prediction, while maintaining orthogonal compatibility with existing methods. Extensive experiments across seven major link prediction benchmarks, including challenging large-scale graphs, demonstrate the benefits of integrating vision into MPNN-based link prediction by adopting the proposed GVN framework.

To our knowledge, GVN is the **first method** that incorporates vision awareness into MPNNs. It demonstrates consistent SOTA effectiveness across various graph link prediction tasks, highlighting a promising but unexplored direction.

**Main Contributions.** (i) We propose novel frameworks GVN and E-GVN, that are the first in incorporating the vision modality into MPNN for link prediction. They are fully compatible with current models, ensuring seamless integrations. (ii) We reveal properties during graph visualization tailored to link prediction tasks and provide good practices. (iii) We analyze the underlying benefits of vision awareness in link prediction and provide empirical demonstrations. (iv) We delve into the scalability on large-scale graphs and propose efficient optimization in E-GVN. (v) We demonstrate the efficacies of GVN and E-GVN across seven common link prediction datasets, where vision awareness brings orthogonal enhancements to the SOTA methods.

## 2. Related Works

**Link Predictor.** Link predictors can be divided into three classes: *1) Node embedding methods* (Perozzi et al., 2014; Tang et al., 2015; Grover & Leskovec, 2016) represent each node as an embedding vector and utilize the embeddings of target nodes to predict links. *2) Link prediction heuristics* (Liben-Nowell & Kleinberg, 2003; Barabási & Albert, 1999; Adamic & Adar, 2003; Zhou et al., 2009) create SFs through manual design. *3) MPNN-based link predictors* produce node representations via the message-passing mechanism (Kipf & Welling, 2017).

**Structural Features.** Structural Features (SFs) are characteristics derived from the graph topology to capture the relationships and properties of nodes and edges. Common SFs are divided into two types: 1) Common Neighbor-based SFs, including the Common Neighbor Count (CN) (Barabási & Albert, 1999), Resource Allocation (RA) (Zhou et al., 2009) and Adamic-Adar (AA) (Adamic & Adar, 2003). 2) Path-based SFs, such as the Shortest Path Distance (SPD) (Dijkstra, 1959), Double Radius Node Labeling (DRNL) (Zhang & Chen, 2018) and Distance Encoding (DE) (Li et al., 2020).

**SF-enhanced MPNNs.** The expressive power of naive MPNN architectures is limited (Zhang et al., 2021), constrained by the 1-dimensional Weisfeiler-Lehman (1-WL) test (Morris et al., 2019), and they fail to finely perceive substructures like triangles (Chen et al., 2020). To overcome these limitations, more recent studies adopt various structural features to enhance MPNNs. For example, SEAL (Zhang & Chen, 2018) incorporates SPD as structural features into MPNNs by concatenating the SPD from each node to the target nodes $u$ and $v$ with the node features. Neo-GNN (Yun et al., 2021) and BUDDY (Chamberlain et al., 2023) use heuristic functions to model high-order common neighbor information. NCNC (Wang et al., 2024) directly concatenates the weighted sum of node representations of common neighbors of $u$ and $v$ with the Hadamard product of MPNN representations of $u$ and $v$.

**Graph Learning with Vision.** Recent research has explored the potential of vision in graph learning. Das et al. (2023) find that the vision-language models have competitive node classification abilities. Wei et al. (2024) shows that visual perception benefits language-based graph reasoning tasks. However, no existing work realizes the potential of vision in enhancing MPNN for link prediction, which is the focus of this paper.

## 3. Preliminaries

**Notations.** A graph $G = (V, E)$ comprises of a set $V$ of $n$ nodes and a set $E$ of $e$ links. We denote the adjacency matrix of $G$ by $A \in \mathbb{R}^{n \times n}$, where $A_{(u,v)} > 0$ if and only if the edge $(u, v)$ exists (i.e., $(u, v) \in E$). We define $N(v) := \{v \in V | A_{uv} > 0\}$ as the set of neighbors of node $v$, and $N_k(v)$ as the set of $k$-hop neighbors of node $v$. In other words, a node $u \in N_k(v)$ if and only if $u$ and $v$ are reachable within $k$ hops. The node feature matrix $\mathbf{X_G} \in \mathbb{R}^{n \times F}$ contains the node features in $G$, where the $v$-th row $\mathbf{x_v}$ corresponds to the feature of node $v$. We use $S_{uv}^k = (V_{uv}^k, E_{uv}^k)$ to denote the $k$-hop subgraph enclosing the link $(u, v)$, where $V_{uv}^k$ is the union of $k$-hop neighbors of $u$ and $v$, and $E_{uv}^k$ is the union of links that can be reached by a $k$-hop walk originating at $u$ or $v$. Moreover, $S_v^k$ is the $k$-hop subgraph starting from node $v$.

**MPNNs for Link Prediction.** The MPNN is a commonly used model for link prediction. In MPNN, message passing is employed to iteratively update node representations based on information exchanged between neighboring nodes. Mathematically, this mechanism can be written as

$$\boldsymbol{h}_v^t = U^t(\boldsymbol{h}_v^{t-1}, \mathbf{AGG}(\{M^t(\boldsymbol{h}_v^{t-1}, \boldsymbol{h}_u^{t-1}) | u \in N(v)\})),$$

$$\mathbf{Y_G} = \mathrm{MPNN}(\mathbf{X_G}, G), \quad \mathbf{y}_v = \boldsymbol{h}_v^k,$$

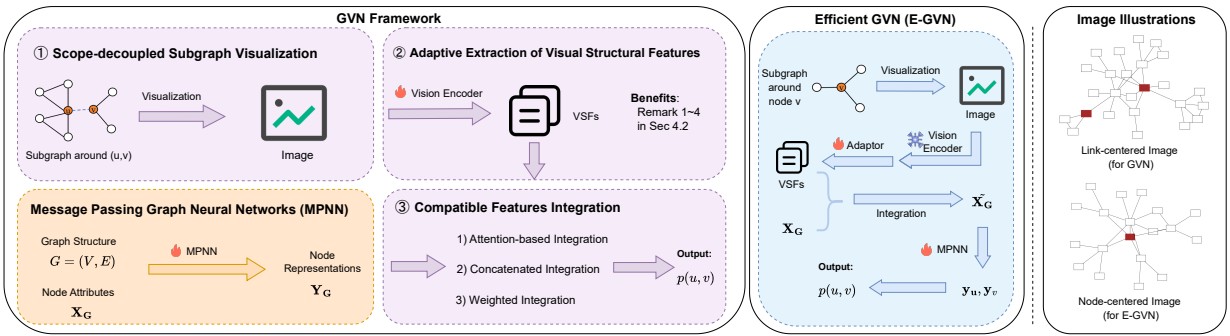

*Figure 1.* An illustration of the GVN framework (left) and its efficient variant E-GVN (middle), with the example illustrations of subgraph visualization images (right). GVN and E-GVN seamlessly incorporate visual awareness of graph structures into MPNNs, which boosts the link prediction capabilities while maintaining the orthogonal compatibility with existing methods.

where $\boldsymbol{h}_v^t$ is the representation of node $v$ after $t$ layers, $U^t$, $M^t$ and $\mathbf{AGG}$ are the update, message-passing, and aggregation functions of layer $t$, respectively. $\mathbf{Y_G} \in \mathbb{R}^{n \times F'}$ is the matrix of final node representations by MPNN for $G$, whose the $v$th row $\mathbf{y}_v$ is the final representation of $v$. Given $\mathbf{Y_G}$, link probabilities can be computed as $p(u,v) = R(\mathbf{y}_u, \mathbf{y}_v)$, where $R$ is a learnable readout function.

## 4. Methodology

In this section, we present the proposed GVN framework. An illustration of GVN is in Figure 1. In Sections 4.1 to 4.3, we will delve into the detailed designs inside, along with addressing the research questions raised in the introduction. In Section 4.4, we propose a computationally efficient variant of GVN for large-scale graphs.

### 4.1. Scope-Decoupled Subgraph Visualization

**Link-Centered Subgraph Visualization.** Beginning with RQ1 (i.e., *For the link prediction task, what is the effective approach to be aware of graph structures from vision*?), we transform graph structures into visual representations by rendering them as images during pre-processing. As large-scale graphs often have thousands of nodes and edges, we utilize $k$-hop subgraph sampling, a common practice in graph learning (Hamilton et al., 2017). This method concentrates on visual awareness via the most relevant local $k$-hop subgraph $S_{uv}^k$ around the central link $(u,v)$, while excluding distant structures that may introduce noise. In line with the locality principle[1] and effective practices of many MPNNs with $k \leq 3$ (Kipf & Welling, 2017; Hamilton et al., 2017; Xu et al., 2019; Velickovic et al., 2018; Chen et al., 2022), we limit $k$, which defines the visualized scope, to a maximum of 3.

---

[1]Locality Principle: Correlated information is contained within the near neighbors (Bronstein et al., 2017; Wu et al., 2020).

To be specific, utilizing existing automated graph visualization tools (Details in Appendix F) such as Graphviz (Gansner & North, 2000), Matplotlib (Tosi, 2009), and Igraph (Gabor Csardi, 2006), a given query link $(u,v)$ and its surrounding $k$-hop subgraph $S_{uv}^k$ are mapped one-to-one into an image $I_{uv}^k$ by a graph visualizer GV. Formally, this is expressed as $I_{uv}^k = \mathrm{GV}(S_{uv}^k, u, v)$.

In this process, three key aspects are noteworthy:

*1) Keeping Style Consistency.* It is crucial that all visualized subgraphs maintain consistency in the graph visualizer GV, such as uniformly using Graphviz with fixed configurations (e.g., graph layout algorithm, node colors, and shapes). This prevents inconsistent image styles from introducing additional learning difficulties and unstable training.

*2) Highlighting the Queried Link.* The unique role of the queried link is essential for link prediction. By default, the end nodes of the queried link are colored to emphasize their roles, and the link itself is masked for prediction.

*3) Ablating Node Labels.* All the node labels are omitted, enabling the model to focus on the graph structure, which benefits generalizability.

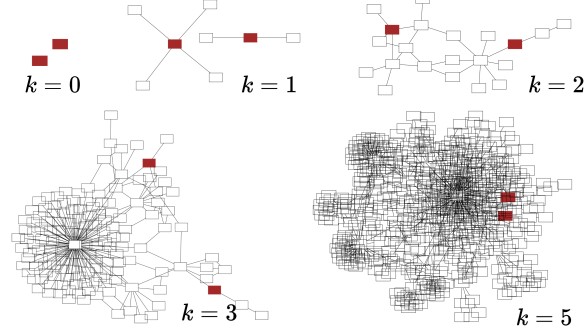

*Figure 2.* Illustrations of visual graph images with different $k$'s.

**Decoupled Vision Scope.** While the perception scope ex-

pands during message passing in MPNN, we fix the visual perception scope $k$ during visualization. This ensures that the scope in vision can be decoupled from which in message passing and offers two key advantages. First, both theoretical analysis and empirical evidence (Zeng et al., 2021) show that a decoupled scope independent from message-passing can improve expressivity of MPNN by addressing the over-smoothing issue (Li et al., 2018; Kenta Oono, 2020; Shen et al., 2024), and enhances the computational scalability of the models by alleviating the neighbor explosion problem (Chiang et al., 2019; Zeng et al., 2019). Second, as shown in Figure 2, an excessively large visual perception scope can lead to overcrowding of nodes/edges on a fixed-size canvas, resulting in a cluttered appearance that complicates graph interpretation. By using a small $k$, we maintain image clarity independent of message passing.

## 4.2. Adaptive Extraction of Visual Structural Features

The obtained visual graph image $I_{uv}^k$ passes through a vision encoder $\text{VE}_\psi$, with trainable parameter $\psi$, to extract visual structural features (VSFs) $\mathbf{v}_{uv} = \text{VE}_\psi(I_{uv}^k) \in \mathbb{R}^S$ for the local subgraph around the queried link $(u, v)$. By default, we use the vision encoder of ResNet50[2], and extract VSFs from its last convolutional layer.

To answer RQ2 (i.e., *Does incorporating vision awareness into graph structures genuinely enhance link prediction? If so, what are the underlying reasons?*), we analyze the superiority of VSFs for link prediction from three aspects:

**Link Discriminative Power.** MPNNs struggle to differentiate links involving isomorphic nodes (Zhang et al., 2021; Chamberlain et al., 2023). As depicted in Figure 3(a), nodes $v_2$ and $v_3$ have isomorphic subgraphs, resulting in identical node representations $\mathbf{y}_{v_2} = \mathbf{y}_{v_3}$ after permutation-equivariant message passing, akin to the 1-dimensional Weisfeiler-Lehman test. Consequently, the link probabilities $p(v_1, v_2) = R(\mathbf{y}_{v_1}, \mathbf{y}_{v_2})$ and $p(v_1, v_3) = R(\mathbf{y}_{v_1}, \mathbf{y}_{v_3})$ are identical. This fails to account for the differences between $v_2$ and $v_3$ w.r.t. $v_1$ and is detrimental for link prediction. For example, $v_2$ is 5 hops away from $v_1$, while $v_3$ is just 2 hops away. Moreover, $v_1$ and $v_3$ share a common neighbor, but not $v_1$ and $v_2$. On the other hand, Figures 3(b) and 3(c) show the 1-hop subgraphs surrounding these two links, and they are clearly very different. Therefore, 1-hop VSFs extracted from them help distinguish these links.

**Remark 4.1.** *With the same perceptive scope, VSFs can discriminate links involving isomorphic nodes, whereas MPNNs or the 1-dimensional Weisfeiler-Lehman test cannot.*

Note that when VSFs provide discriminative power in its

[2]https://download.pytorch.org/models/resnet50-0676ba61.pth.

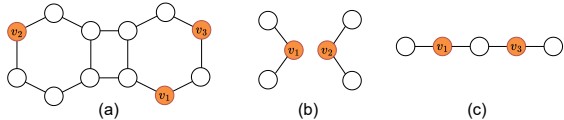

*Figure 3.* (a) An illustration of the challenge in distinguishing links with isomorphic nodes. (b) and (c) are the 1-hop subgraphs surrounding the links $(v_1, v_2)$ and $(v_1, v_3)$, respectively.

scope, our method allows the model to set the independent scope of MPNNs via its depth. This enables the model to benefit both from the refined structure awareness provided by VSFs and the capability to access more structure information in a wider scope through deep MPNNs.

**Fine-grained Substructure Awareness.** The ability to capture substructures (motifs) is another perspective of MPNN expressive power and plays a crucial role in areas like biology, molecular, and social networks (Chen et al., 2020; Kanatsoulis & Ribeiro, 2024; Yan et al., 2024). However, this can be challenging for MPNNs (Chen et al., 2020; Arvind et al., 2020). We demonstrate in the following that VSFs are helpful for MPNNs in capturing substructures. Specifically, we follow the experimental setup in (Chen et al., 2020). This involves generating two types of graph datasets, Erdős-Rényi graphs and random regular graphs, and compute the counts of two substructures including triangles and 3-stars as ground-truth labels. We use MPNN followed by a 3-layer MLP decoder as the architecture and the VSFs are concatenated after the MPNN-produced representations. The evaluation metric is the normalized mean squared error (i.e., ratio of the mean squared error over the variance of ground-truth labels). Table 1 shows the best and median (third-best) performance over five runs with different random seeds. As can be seen, incorporating VSFs significantly improves the substructure capturing abilities of MPNNs. More details of this experiment are provided in Appendix E.

| | Erdős-Rényi | | | | Random Regular | | | |
| | Triangle | | 3-Star | | Triangle | | 3-Star | |
| | Best | Median | Best | Median | Best | Median | Best | Median |
|---|---|---|---|---|---|---|---|---|
| GCN | 0.69 | 0.83 | 0.49 | 0.55 | 1.84 | 2.79 | 2.81 | 4.67 |
| VSF+GCN | 6.76E-9 | 1.24E-8 | 3.22E-8 | 1.75E-7 | 2.06E-7 | 4.97E-7 | 7.11E-8 | 1.56E-7 |
| SAGE | 0.13 | 0.17 | 2.35E-6 | 1.92E-5 | 0.37 | 0.50 | 4.94E-7 | 7.72E-5 |
| VSF+SAGE | 3.24E-6 | 2.70E-5 | 1.27E-8 | 9.91E-7 | 1.02E-5 | 4.39E-5 | 5.11E-8 | 4.04E-6 |

*Table 1.* The performance of MPNNs with VSFs for counting triangles and 3-star on Erdős-Rényi and Random regular graphs.

**Remark 4.2.** *VSFs can provide fine-grained substructure capturing ability for MPNNs via VSFs.*

**Information-rich and Adaptive SF Extraction.** While the design of existing structural features rely on heuristics, VSFs encode the whole subgraph, making them more information-rich. Figure 4 shows the reproduction ratios[3]

[3]The reproduction ratio is defined as the percentage (%) of SFs successfully predicted.

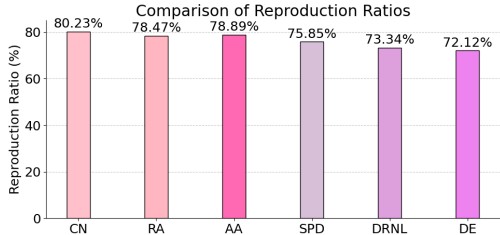

*Figure 4.* Reproduction ratios of various structural features based on VSFs on the *ogbl-ddi* dataset.

of various SFs based on VSFs on *ogbl-ddi*, by feeding the VSFs extracted from the pretrained ResNet50 into a trainable 3-layer MLP. As can be seen, VSFs effectively capture most of the information from a wide array of structural features, underscoring their comprehensive nature as a versatile structural feature pool.

**Remark 4.3.** *VSFs involve the information of a wide range of structural features.*

The reproduction ratios of various SFs reflect the relative distribution of information within the VSFs. Table 2 compares the reproduction ratios of SFs between VSFs obtained from a pre-trained ResNet50 and those after finetuning. The latter is derived from a ResNet50 encoder fine-tuned for link prediction with a 2-layer MLP on the *ogbl-ddi* and *PubMed* datasets. Note that *ogbl-ddi* is a very dense graph, while *PubMed* is much sparser. As shown in Table 2, after finetuning on *ogbl-ddi*, the reproduction ratio of path-based SFs (SPD, DRNL, and DE) consistently decreases. This aligns with the under-performance observed in path-based link prediction models like SEAL (Zhang & Chen, 2018) and NBFNet (Zhu et al., 2021) on the *ogbl-ddi* dataset. This is because in the *ogbl-ddi* dataset, most node pairs are reachable within two hops, making path-based structural features like SPD less informative. On the other hand, in the sparser *PubMed* dataset where path information is more valuable, the reproduction ratios of common neighbor-based similarity functions (CN, RA, AA) consistently increase after finetuning. This trend is demonstrated by models such as NeoGNN, Buddy, and NCCN (Yun et al., 2021; Chamberlain et al., 2023; Wang et al., 2024), which consistently benefit from common neighbor similarity functions across various datasets. These observations highlight the ability of VSFs to adapt the internal information types to meet the specific demands of different scenarios through finetuning.

| | Common Neighbor-based SFs | | | Path-based SFs | | |
|---|---|---|---|---|---|---|
| | CN | RA | AA | SPD | DRNL | DE |
| *ogbl-ddi (w/o)* | 80.23% | 78.47% | 78.89% | 75.85% | 73.34% | 72.12% |
| *ogbl-ddi (w/)* | 82.39% | 79.66% | 80.24% | 31.11% | 29.20% | 35.49% |
| *PubMed (w/o)* | 76.39% | 78.87% | 72.56% | 76.60% | 72.36% | 71.19% |
| *PubMed (w/)* | 83.96% | 82.80% | 78.86% | 78.77% | 73.34% | 71.59% |

*Table 2.* Reproduction ratios of SFs in VSFs with and without finetuning on the *ogbl-ddi* and *PubMed* datasets.

**Remark 4.4.** *Through finetuning, VSFs can adaptively ad-*

*just their emphasis on different types of structural information, thus providing scenario adaptability.*

VSFs are not permutation-equivariant, yet they remain effective with MPNNs, as will be shown in Section 5.1. This is potentially attributed to two reasons: 1) Learnable VSFs capture specific information like structural patterns or motifs (see Remark 4.2). 2) Visualization introduces permutation-based augmentations (e.g., varying layouts), making the decoder less sensitive to node order.

### 4.3. Compatible Features Integration

After extracting valuable visual structure features $\mathbf{v}_{uv}$, the next step is to integrate them with MPNNs, addressing RQ3 (i.e., *If vision-based structural awareness proves beneficial, how can it be effectively fused into the MPNN architecture?*).

To ensure general applicability of the GVN framework, the feature integration process should be compatible with existing methods, to benefit from both previous advances and the newly introduced visual structural awareness. Based on this compatibility principle, we offer three widely applicable feature integration strategies, all of which are independent of the message-passing process in MPNNs.

*1) Attention-based Integration.* We first project $\mathbf{v}_{uv} \in \mathbb{R}^S$ to $\tilde{\mathbf{v}}_{uv} \in \mathbb{R}^{F'}$ for dimension alignment. Then the MPNN-produced node representations $\mathbf{y}_u \in \mathbb{R}^{F'}$ and $\mathbf{y}_v \in \mathbb{R}^{F'}$ are updated with $\tilde{\mathbf{v}}_{uv}$ via cross-attention (denoted CA), resulting in vision-aware node representations $\tilde{\mathbf{y}}_u = \text{CA}(\mathbf{y}_u, \tilde{\mathbf{v}}_{uv})$ and $\tilde{\mathbf{y}}_v = \text{CA}(\mathbf{y}_v, \tilde{\mathbf{v}}_{uv})$. Finally the link existence probability can be computed as $p(u, v) = R(\tilde{\mathbf{y}}_u, \tilde{\mathbf{y}}_v)$, where $R$ is the readout function.

*2) Concatenated Integration.* We concatenate $\mathbf{v}_{uv}$ with $\mathbf{y}_u$ and $\mathbf{y}_v$ to produce vision-aware node representations $\tilde{\mathbf{y}}_u$ and $\tilde{\mathbf{y}}_v$, as: $\tilde{\mathbf{y}}_u = \mathbf{y}_u || \mathbf{v}_{uv}, \quad \tilde{\mathbf{y}}_v = \mathbf{y}_v || \mathbf{v}_{uv}$. Then we use the readout function to produce the link existence probability $p(u, v) = R(\tilde{\mathbf{y}}_u, \tilde{\mathbf{y}}_v)$.

*3) Weighted Integration.* We use an MLP to construct a vision-based decoder model (denoted VDecoder), which takes $\mathbf{v}_{uv}$ as input to directly predict the link existence probability based solely on visual structural awareness, i.e., $p_{vision}(u, v) = \text{VDecoder}(\mathbf{v}_{uv})$. Simultaneously, the readout function predicts the link existence probability based on the node representations produced by the MPNN, i.e., $p_{MPNN}(u, v) = R(\mathbf{y}_u, \mathbf{y}_v)$. The final link existence probability $p(u, v)$ is a weighted integration of both predictions with a learnable weight $\delta$, i.e., $p_{(u,v)} = \delta \cdot p_{vision}(u, v) + (1 - \delta) \cdot p_{MPNN}(u, v)$.

These integration strategies can be freely chosen as needed. Empirically, we observe that attention-based integration is more effective and therefore used by default.

Thus far, we have presented the GVN framework to address

**Algorithm 1** The GVN Framework (Attention-based).

**Require:** Graph $G = (V, E)$ with node attributes $\mathbf{X}_G$, queried links $\mathcal{Q} = \{(u, v)\}$.
**Ensure:** Link existence probabilities $\{p(u, v) | (u, v) \in \mathcal{Q}\}$
 1: $\mathbf{Y}_G \leftarrow \text{MPNN}(G, \mathbf{X}_G)$
 2: **for** each $(u, v) \in \mathcal{Q}$ **do**
 3:   **Step 1**: Scope-decoupled Subgraph Visualization
 4:     $S_{uv}^k \leftarrow$ Extract $k$-hop subgraph around $(u, v)$
 5:     $I_{uv}^k \leftarrow \text{GV}(S_{uv}^k)$
 6:   **Step 2**: Adaptive Extraction of VSFs
 7:     $\mathbf{v}_{uv} \leftarrow \text{VE}_\psi(I_{uv}^k)$
 8:   **Step 3**: Compatible Features Integration (default)
 9:     $\mathbf{y}_u, \mathbf{y}_v \leftarrow \mathbf{Y}_G[u], \mathbf{Y}_G[v]$
10:     $\tilde{\mathbf{v}}_{uv} \leftarrow \text{Linear}(\mathbf{v}_{uv})$
11:     $\tilde{\mathbf{y}}_u, \tilde{\mathbf{y}}_v \leftarrow \text{CA}(\mathbf{y}_u, \tilde{\mathbf{v}}_{uv}), \text{CA}(\mathbf{y}_v, \tilde{\mathbf{v}}_{uv})$
12:     $p(u, v) \leftarrow R(\tilde{\mathbf{y}}_u, \tilde{\mathbf{y}}_v)$
13:   Output $p(u, v)$
14: **end for**

the three research questions corresponding to the three main steps in GVN to integrate vision information into MPNNs. We summarize the GVN framework with default attention-based integration in Algorithm 1. GVN Algorithms with other integration are in Appendix A.1.

### 4.4. E-GVN: An Efficient Variant

Considering the scalability of GVN on large-scale graphs, we propose a more efficient variant called **Efficient GVN (E-GVN)**, whose architecture is presented in Figure 1 and the algorithm is in Appendix A.2. The specialized modifications are introduced as follows.

**Node-centered Subgraph Visualization.** GVN requires subgraph visualization for the surrounding area of each link, which can be time-consuming. Therefore, in E-GVN, the visualization area is changed from a link-centered $k$-hop neighborhood $S_{uv}^k$ to a node-centered $k$-hop subgraph $S_v^k$, with the central node $v$ colored to highlight its role. This visualization process is formalized as $I_v^k = \text{GV}(S_v^k, v)$. Due to the fact that the number of nodes is much smaller than the number of links in most natural graphs, this approach significantly reduces the visualization cost from $O(l)$ to $O(n)$, where $l$ and $n$ denote the total number of links and nodes.

**Partial-adaptive VSFs.** GVN use end-to-end finetuning of vision encoder $\text{VE}_\psi$ to refine the VSFs. However, such end-to-end finetuning requires retaining all visualization images and thus lead to substantial memory overhead. To enhance scalability in terms of space, E-GVN first freezes the vision encoder to extract static VSF $\mathbf{v}_v$. Instead, those static VSFs $\{\mathbf{v}_v\}$ can be stored and loaded as a fixed vector repository, significantly reducing the storage payload. After that, as we still want the VSFs to flexibly adapt to the different scenarios, we additionally append a trainable Adapter with

learnable parameter $\phi$, which processes the static VSF $\mathbf{v}_v$ to adaptive $\tilde{\mathbf{v}}_v$, where $\tilde{\mathbf{v}}_v = \text{Adaptor}_\phi(\mathbf{v}_v)$ for each node $v$.

**Pre-MPNN Integration on Attributes.** Unlike GVN, which integrates VSFs after the MPNN, E-GVN incorporates $\tilde{\mathbf{v}}_v$ into the node attribute $\mathbf{x}_v$ to produce vision-aware node attribute $\tilde{\mathbf{x}}_v$, which are then fed into the MPNN as inputs. Due to the integration occurs before message passing, this allows the vision-aware structural information to participate in the message propagation and aggregation to get further enhanced.

To achieve this, the three integration strategies in GVN are adapted as follows: 1) Attention-based Integration: $\tilde{\mathbf{x}}_v = \text{CA}(\mathbf{x}_v, \tilde{\mathbf{v}}_v)$; 2) Concatenated Integration: $\tilde{\mathbf{x}}_v = \mathbf{x}_v || \tilde{\mathbf{v}}_v$; 3) Weighted Integration: $\tilde{\mathbf{x}}_v = \delta \text{Linear}_{\phi_1}(\mathbf{x}_v) + (1 - \delta)\text{Linear}_{\phi_2}(\tilde{\mathbf{v}}_v)$, where $\delta$ is a learnable weight and $\text{Linear}_{\phi_1}, \text{Linear}_{\phi_2}$ are linear layers. Finally, the MPNN-produced node representations are processed by a readout function to predict the link existence probability.

**Time Complexity.** Let $n$ and $l$ be the numbers of nodes and links, respectively, $F$ and $F'$ be the dimensions of node features and MPNN-produced representations, respectively, $O(\text{MPNN})$ and $O(\text{VE})$ be the complexities of the MPNN and vision encoder, respectively, and $S$ be the dimension of the VSF. With the default attention-based integration, the time complexities of GVN and E-GVN are $O(\text{MPNN}) + lO(\text{VE}) + O(l(SF' + F'^2))$ and $O(\text{MPNN}) + nO(\text{VE}) + O(n(S^2 + SF + F^2))$, respectively. We can see that E-GVN reduces the computational complexity, as $n$ is much less than $l$. More details are provided in Appendix B.

## 5. Experiments

In this section, we conduct a series of experiments to demonstrate the effectiveness of the proposed GVN and E-GVN.

### 5.1. Evaluation on Real-World Datasets

**Datasets.** We conduct experiments on seven widely used link prediction benchmarking, including the Planetoid citation networks: *Cora* (McCallum et al., 2000), *Citeseer* (Sen et al., 2008), and *Pubmed* (Namata et al., 2012), and the large-scale OGB link prediction datasets (Hu et al., 2020): *ogbl-collab*, *ogbl-ppa*, *ogbl-citation2* and *ogbl-ddi*. Statistics of these datasets are shown in Appendix C.

**Baselines.** Baseline methods used include 1) Link prediction heuristics: Common Neighbor counts (CN) (Barabási & Albert, 1999), Adamic-Adar (AA) (Adamic & Adar, 2003), and Resource Allocation (RA) (Zhou et al., 2009); 2) MPNNs: GraphSAGE (Hamilton et al., 2017) and Graph Convolutional Network (GCN) (Kipf & Welling, 2017); 3) Advanced SF-enhanced MPNNs: SEAL (Zhang & Chen, 2018) and NBFNet (Zhu et al., 2021) with path-based SFs,

*Table 3.* Link prediction performance (average score ± standard deviation). "-" indicates that the training time is $> 12$ hour/epoch (for GVN) or out of memory (for NBFnet). The best performance is shown in bold, and the second-best is underlined.

| | *Cora* (HR@100) | *Citeseer* (HR@100) | *Pubmed* (HR@100) | *ogbl-collab* (HR@50) | *ogbl-ppa* (HR@100) | *ogbl-citation2* (MRR) | *ogbl-ddi* (HR@20) |
|---|---|---|---|---|---|---|---|
| CN | $33.92_{\pm0.46}$ | $29.79_{\pm0.90}$ | $23.13_{\pm0.15}$ | $56.44_{\pm0.00}$ | $27.65_{\pm0.00}$ | $51.47_{\pm0.00}$ | $17.73_{\pm0.00}$ |
| AA | $39.85_{\pm1.34}$ | $35.19_{\pm1.33}$ | $27.38_{\pm0.11}$ | $64.35_{\pm0.00}$ | $32.45_{\pm0.00}$ | $51.89_{\pm0.00}$ | $18.61_{\pm0.00}$ |
| RA | $41.07_{\pm0.48}$ | $33.56_{\pm0.17}$ | $27.03_{\pm0.35}$ | $64.00_{\pm0.00}$ | $49.33_{\pm0.00}$ | $51.98_{\pm0.00}$ | $27.60_{\pm0.00}$ |
| SAGE | $55.02_{\pm4.03}$ | $57.01_{\pm3.74}$ | $39.66_{\pm0.72}$ | $48.10_{\pm0.81}$ | $16.55_{\pm2.40}$ | $82.60_{\pm0.36}$ | $53.90_{\pm4.74}$ |
| GCN | $66.79_{\pm1.65}$ | $67.08_{\pm2.94}$ | $53.02_{\pm1.39}$ | $44.75_{\pm1.07}$ | $18.67_{\pm1.32}$ | $84.74_{\pm0.21}$ | $37.07_{\pm5.07}$ |
| $\text{GVN}_{GCN}$ | $81.13_{\pm0.86}$ | $83.93_{\pm0.97}$ | $73.17_{\pm1.02}$ | - | - | - | - |
| $\text{E-GVN}_{GCN}$ | $80.01_{\pm1.55}$ | $82.85_{\pm1.90}$ | $71.94_{\pm1.37}$ | $62.14_{\pm1.37}$ | $32.15_{\pm1.58}$ | $86.10_{\pm0.13}$ | $60.21_{\pm6.67}$ |
| Neo-GNN | $80.42_{\pm1.31}$ | $84.67_{\pm2.16}$ | $73.93_{\pm1.19}$ | $57.52_{\pm0.37}$ | $49.13_{\pm0.60}$ | $87.26_{\pm0.84}$ | $63.57_{\pm3.52}$ |
| SEAL | $81.71_{\pm1.30}$ | $83.89_{\pm2.15}$ | $75.54_{\pm1.32}$ | $64.74_{\pm0.43}$ | $48.80_{\pm3.16}$ | $87.67_{\pm0.32}$ | $30.56_{\pm3.86}$ |
| NBFnet | $71.65_{\pm2.27}$ | $74.07_{\pm1.75}$ | $58.73_{\pm1.99}$ | - | - | - | $4.00_{\pm0.58}$ |
| BUDDY | $88.00_{\pm0.44}$ | $92.93_{\pm0.27}$ | $74.10_{\pm0.78}$ | $65.94_{\pm0.58}$ | $49.85_{\pm0.20}$ | $87.56_{\pm0.11}$ | $78.51_{\pm1.36}$ |
| NCNC | $89.65_{\pm1.36}$ | $93.47_{\pm0.95}$ | $81.29_{\pm0.95}$ | $\underline{66.61}_{\pm0.71}$ | $\underline{61.42}_{\pm0.73}$ | $\underline{89.12}_{\pm0.40}$ | $\underline{84.11}_{\pm3.67}$ |
| $\text{GVN}_{NCNC}$ | $\underline{90.70}_{\pm0.56}$ | $\underline{94.12}_{\pm0.58}$ | $\underline{82.17}_{\pm0.77}$ | - | - | - | - |
| $\text{E-GVN}_{NCNC}$ | $\mathbf{91.47}_{\pm0.36}$ | $\mathbf{94.44}_{\pm0.53}$ | $\mathbf{84.02}_{\pm0.55}$ | $\mathbf{68.14}_{\pm0.75}$ | $\mathbf{63.45}_{\pm0.66}$ | $\mathbf{90.72}_{\pm0.24}$ | $\mathbf{87.31}_{\pm3.04}$ |

Neo-GNN (Yun et al., 2021), BUDDY (Chamberlain et al., 2023), and NCNC (Wang et al., 2024) that is the SOTA SF-enhanced MPNN.

**Performance Evaluation.** The use of evaluation metrics follows (Chamberlain et al., 2023; Wang et al., 2024). Specifically, for the Planetoid datasets, we use the hit-ratio at 100 (HR@100), while for the OGB datasets, we use the metrics in their official documents,[4] i.e., hit-ratio at 50 (HR@50) for *ogbl-collab*, HR@100 for *ogbl-ppa*, Mean Reciprocal Rank (MRR) for *ogbl-citation2*, and hit-ratio at 20 (HR@20) for *ogbl-ddi*.[5] All results are reported as average and standard variation over 10 trials with different random seeds.

**Implementation Details.** All experiments are conducted on an NVIDIA A100 80G GPU. For GVN and E-GVN, the candidate hyperparameters include the visual perception scope (i.e., the subgraph visualization hop) $k$ ranging from 1 to 3, the hidden dimension ranging from 512 to 2048, the number of MPNN layers and readout predictor layers varying from 1 to 3, the two separate learning rates for trainable VSFs and compatible feature integration among $\{10^{-7}, 10^{-5}, 0.001, 0.01\}$, and the weight decay from 0 to 0.0001. The hyperparameters with the best validation accuracy are selected. For the model parameters, we utilize the Adam algorithm (Kingma, 2014) as the optimizer. As default settings, we use Graphviz (Gansner & North, 2000) as the graph visualizer and use a pretrained ResNet50 (He et al., 2016) as the vision encoder. More details on the experimental setup are in Appendix D.

**Configurations of the Proposed Methods.** We use four configurations of the GVN and E-GVN frameworks: $\text{GVN}_{GCN}$, $\text{GVN}_{NCNC}$, $\text{E-GVN}_{GCN}$, and $\text{E-GVN}_{NCNC}$, as shown in Table 3. The subscript (i.e., 'GCN' or 'NCNC')

indicates the MPNN model used within the frameworks. Our choice of MPNNs is based on the following considerations: First, using GCN, a classic and straightforward MPNN, within the proposed frameworks directly demonstrates the enhancement provided by vision awareness when compared to MPNN alone. This allows us to clearly see the impact of incorporating the visual modality into MPNNs. Second, based on NCNC, a state-of-the-art MPNN enhanced by advanced structural features, the comparison between $\text{GVN}_{NCNC}$/$\text{E-GVN}_{NCNC}$ and NCNC will highlight the additional performance improvements from the vision awareness that are orthogonal to existing advanced methods. This demonstrates how visual features can complement and enhance existing sophisticated models. Although we focus on these two MPNNs, the proposed GVN and E-GVN are compatible with most MPNN models.

**Main Results.** Table 3 compares the performance of the proposed methods with various baselines.

(i) By integrating visual structural features into GCN, GVN enhances GCN with relative improvements on link prediction performance (HR@100) by 21.47%, 25.13%, and 38.00% on *Cora*, *Citeseer* and *PubMed*, respectively. Similarly, E-GVN achieves relative improvements of 19.79%, 23.51%, and 35.68% upon GCN on these three datasets. Additionally, due to the efficient designs, E-GVN extends the visual benefits to large-scale graph datasets *ogbl-collab*, *ogbl-ppa*, *ogbl-citation2* and *ogbl-ddi*, bringing relative improvements of 38.86% (HR@50), 72.20% (HR@100), 1.60% (MRR), and 62.42% (HR@20), respectively. Therefore, consistent results indicate that visual awareness has been effectively integrated into GCN by the proposed methods and significantly enhances the link prediction performance of GCN.

(ii) Based on SF-enhanced NCNC model, both $\text{GVN}_{NCNC}$ and $\text{E-GVN}_{NCNC}$ still show remarkable improvements.

---

[4]https://ogb.stanford.edu/docs/leader_linkprop/.
[5]Evaluation results on other metrics are in Section 5.2.

*Table 4.* The performance comparison of the proposed methods against the strongest baseline NCNC across broader metrics.

| | | Cora | Citeseer | Pubmed | Collab | PPA | Citation2 | DDI |
|---|---|---|---|---|---|---|---|---|
| hit@1 | $\text{GVN}_{NCNC}$ | $\mathbf{11.75}_{\pm \mathbf{7.72}}$ | $51.69_{\pm 7.91}$ | $\mathbf{18.66}_{\pm \mathbf{8.85}}$ | - | - | - | - |
| | $\text{E-GVN}_{NCNC}$ | $8.66_{\pm 4.39}$ | $\mathbf{59.30}_{\pm \mathbf{5.53}}$ | $16.88_{\pm 9.58}$ | $\mathbf{11.04}_{\pm \mathbf{3.01}}$ | $6.53_{\pm 1.52}$ | $\mathbf{86.62}_{\pm \mathbf{1.04}}$ | $\mathbf{0.42}_{\pm \mathbf{0.08}}$ |
| | NCNC | $10.90_{\pm 11.40}$ | $32.45_{\pm 17.01}$ | $8.57_{\pm 6.76}$ | $9.82_{\pm 2.49}$ | $7.78_{\pm 0.36}$ | $84.66_{\pm 1.15}$ | $0.16_{\pm 0.07}$ |
| hit@3 | $\text{GVN}_{NCNC}$ | $26.66_{\pm 5.96}$ | $59.97_{\pm 6.21}$ | $\mathbf{32.23}_{\pm \mathbf{5.69}}$ | - | - | - | - |
| | $\text{E-GVN}_{NCNC}$ | $\mathbf{27.55}_{\pm \mathbf{6.37}}$ | $\mathbf{66.76}_{\pm \mathbf{4.20}}$ | $31.21_{\pm 5.98}$ | $\mathbf{26.31}_{\pm \mathbf{7.74}}$ | $\mathbf{18.88}_{\pm \mathbf{1.21}}$ | $\mathbf{94.29}_{\pm \mathbf{0.96}}$ | $\mathbf{2.12}_{\pm \mathbf{0.33}}$ |
| | NCNC | $25.04_{\pm 11.40}$ | $50.49_{\pm 12.01}$ | $17.58_{\pm 6.57}$ | $21.07_{\pm 5.46}$ | $16.58_{\pm 0.60}$ | $92.37_{\pm 0.56}$ | $0.59_{\pm 0.42}$ |
| hit@10 | $\text{GVN}_{NCNC}$ | $\mathbf{58.83}_{\pm \mathbf{5.29}}$ | $75.28_{\pm 3.03}$ | $40.34_{\pm 2.28}$ | - | - | - | - |
| | $\text{E-GVN}_{NCNC}$ | $55.98_{\pm 4.14}$ | $\mathbf{77.12}_{\pm \mathbf{2.95}}$ | $\mathbf{47.90}_{\pm \mathbf{2.86}}$ | $43.12_{\pm 5.77}$ | $\mathbf{31.16}_{\pm \mathbf{1.67}}$ | $\mathbf{97.07}_{\pm \mathbf{1.01}}$ | $\mathbf{50.88}_{\pm \mathbf{11.35}}$ |
| | NCNC | $53.78_{\pm 7.33}$ | $69.59_{\pm 4.48}$ | $34.29_{\pm 4.43}$ | $\mathbf{43.22}_{\pm \mathbf{6.19}}$ | $26.67_{\pm 1.51}$ | $96.99_{\pm 0.64}$ | $45.64_{\pm 14.12}$ |
| hit@20 | $\text{GVN}_{NCNC}$ | $\mathbf{70.01}_{\pm \mathbf{4.44}}$ | $81.11_{\pm 1.30}$ | $53.33_{\pm 2.67}$ | - | - | - | - |
| | $\text{E-GVN}_{NCNC}$ | $69.55_{\pm 3.46}$ | $\mathbf{82.02}_{\pm \mathbf{1.46}}$ | $\mathbf{56.92}_{\pm \mathbf{2.33}}$ | $56.87_{\pm 2.97}$ | $\mathbf{44.06}_{\pm \mathbf{2.03}}$ | $\mathbf{98.17}_{\pm \mathbf{0.97}}$ | $\mathbf{87.31}_{\pm \mathbf{3.04}}$ |
| | NCNC | $67.10_{\pm 2.96}$ | $79.05_{\pm 2.68}$ | $51.42_{\pm 3.81}$ | $\mathbf{57.83}_{\pm \mathbf{3.14}}$ | $35.00_{\pm 2.22}$ | $97.22_{\pm 0.94}$ | $83.92_{\pm 3.25}$ |
| hit@50 | $\text{GVN}_{NCNC}$ | $82.06_{\pm 1.94}$ | $88.88_{\pm 0.98}$ | $\mathbf{71.66}_{\pm \mathbf{2.75}}$ | - | - | - | - |
| | $\text{E-GVN}_{NCNC}$ | $\mathbf{82.99}_{\pm \mathbf{2.95}}$ | $\mathbf{88.97}_{\pm \mathbf{0.58}}$ | $71.55_{\pm 1.19}$ | $\mathbf{68.14}_{\pm \mathbf{0.75}}$ | $\mathbf{52.58}_{\pm \mathbf{0.30}}$ | $\mathbf{99.09}_{\pm \mathbf{0.66}}$ | $\mathbf{95.95}_{\pm \mathbf{0.75}}$ |
| | NCNC | $81.36_{\pm 1.86}$ | $88.60_{\pm 1.51}$ | $69.25_{\pm 2.87}$ | $66.88_{\pm 0.66}$ | $48.66_{\pm 0.18}$ | $99.01_{\pm 0.53}$ | $94.85_{\pm 0.56}$ |
| hit@100 | $\text{GVN}_{NCNC}$ | $90.70_{\pm 0.56}$ | $94.12_{\pm 0.58}$ | $82.17_{\pm 0.77}$ | - | - | - | - |
| | $\text{E-GVN}_{NCNC}$ | $\mathbf{91.47}_{\pm \mathbf{0.36}}$ | $\mathbf{94.44}_{\pm \mathbf{0.53}}$ | $\mathbf{84.02}_{\pm \mathbf{0.55}}$ | $70.83_{\pm 2.25}$ | $\mathbf{63.45}_{\pm \mathbf{0.66}}$ | $\mathbf{99.51}_{\pm \mathbf{0.39}}$ | $\mathbf{97.99}_{\pm \mathbf{0.27}}$ |
| | NCNC | $89.05_{\pm 1.24}$ | $93.13_{\pm 1.13}$ | $81.18_{\pm 1.24}$ | $\mathbf{71.96}_{\pm \mathbf{0.14}}$ | $62.02_{\pm 0.74}$ | $99.37_{\pm 0.27}$ | $97.60_{\pm 0.22}$ |
| mrr | $\text{GVN}_{NCNC}$ | $\mathbf{24.66}_{\pm \mathbf{4.51}}$ | $62.74_{\pm 6.63}$ | $26.32_{\pm 6.67}$ | - | - | - | - |
| | $\text{E-GVN}_{NCNC}$ | $23.27_{\pm 3.39}$ | $\mathbf{66.49}_{\pm \mathbf{3.53}}$ | $\mathbf{27.11}_{\pm \mathbf{5.88}}$ | $\mathbf{18.04}_{\pm \mathbf{3.01}}$ | $\mathbf{19.66}_{\pm \mathbf{0.11}}$ | $\mathbf{90.72}_{\pm \mathbf{0.24}}$ | $\mathbf{13.32}_{\pm \mathbf{2.75}}$ |
| | NCNC | $23.55_{\pm 9.67}$ | $45.64_{\pm 11.78}$ | $15.63_{\pm 4.13}$ | $17.68_{\pm 2.70}$ | $14.37_{\pm 0.06}$ | $89.12_{\pm 0.40}$ | $8.61_{\pm 1.37}$ |

Although NCNC is a leading link prediction model by enhancing MPNN with dedicated common neighbor structural features, adopting GVN and E-GVN brings additional improvements over NCNC. These results demonstrate that the visual awareness provided by GVN or E-GVN own advantages not covered by existing methods and can be used in conjunction with current SOTA methods for better performance. Therefore, $\text{GVN}_{NCNC}$ and $\text{E-GVN}_{NCNC}$ surpass all baselines and are established as more powerful link prediction approaches.

(iii) Compared to GVN, the computationally efficient design of E-GVN makes it more suitable for large-scale graphs. $\text{GVN}_{GCN}$ is more powerful than $\text{E-GVN}_{GCN}$. This may be because the visual structural awareness in GVN is link-based, therefore explicitly revealing link pairwise relations such as common neighbors or shortest path differences for distinguishing links with isomorphic nodes better. However, when using SF-enhanced NCNC, the improvements brought by $\text{E-GVN}_{NCNC}$ surpass those of $\text{GVN}_{NCNC}$. We attribute this to the fact that in $\text{GVN}_{NCNC}$, both the link-centered visual awareness and the common-neighbor-based SF can reveal the common neighbor information between the link nodes, which makes their effects partially overlap. In contrast, the pre-MPNN integration in E-GVN allows vision-aware structural information to be further aggregated during message passing, which provides more comprehensive visual awareness.

### 5.2. Evaluation on Broader Metrics

Besides the metrics listed in Table 3, the performances of proposed methods against the strongest baseline NCNC with broader metrics are presented in Table 4. In total, E-

GVN achieves 39 best scores (in bold), GVN achieves 7 best scores, and our strongest baseline NCNC achieves 3 best scores. Therefore, $\text{GVN}_{NCNC}$ and $\text{E-GVN}_{NCNC}$ consistently outperform NCNC in more comprehensive metrics.

### 5.3. Ablation and Sensitivity Analysis

**Visualization Style Consistency.** Table 5 shows that inconsistent styles degrade performance and emphasize keeping styles consistent. The inconsistent styles are sampled from a fixed range of node colors, shapes, and graph visualizers, as detailed in Appendix G. In Appendix H, we include more comprehensive findings by studying the impacts of node colors (Appendix H.1), node shapes (Appendix H.2), and node labeling strategies (Appendix H.3).

**Visualization Scope Impact.** Table 6 shows the performance with varying visualization scopes $k$. The results consistently show that $k = 2$ is enough for effective VSFs.

**Graph Visualizer Selection.** Table 7 shows the performance with different graph visualizers. The results demonstrate stable performance across various graph visualizers.

**Vision Encoder Selection.** Table 8 shows the performance with different vision encoders. The results show stable performance across various vision encoders.

**Feature Integration Strategies.** Table 9 shows the performance when varying feature integration strategies in Section 4.3. Experimental results suggest that in most cases, the default attention-based integration (denoted "Attention" in Table 9) is more effective than concatenated integration (denoted "Concat") and weighted integration (denoted "Weight").

**VSF Adaptivity.** Table 10 shows the impacts of VSF Adaptivity, where "freeze" means using a frozen vision encoder, making VSFs static, "partial" adopts the partial adaptive VSFs proposed in Section 4.4 that freezes the vision encoder but supplements it with a learnable adapter to provide some adaptivity, and "full" means the models are fully fine-tuned. Results show that static VSFs lose adaptivity, leading to performance degradation. The "partial" strategy can provide comparable performance with the "full" strategy, demonstrating its effectiveness.

*Table 5.* Performance comparison between consistent and inconsistent visualization (HR@100).

| | *Cora* | | *Citeseer* | |
| | consistent | inconsistent | consistent | inconsistent |
|---|---|---|---|---|
| $GVN_{NCNC}$ | **90.70**±0.56 | 89.88±0.47 | **94.12**±0.58 | 93.65±0.64 |
| $E\text{-}GVN_{NCNC}$ | **91.47**±0.36 | 90.29±0.61 | **94.44**±0.53 | 93.99±0.51 |

*Table 6.* Performance comparison of the visualization scopes $k$.

| | *Cora* | | | *Citeseer* | | |
| | $k=1$ | $k=2$ | $k=3$ | $k=1$ | $k=2$ | $k=3$ |
|---|---|---|---|---|---|---|
| $GVN_{NCNC}$ | 89.76±0.78 | **90.70**±0.56 | 89.68±0.97 | 92.51±0.86 | **94.12**±0.58 | 92.33±0.91 |
| $E\text{-}GVN_{NCNC}$ | 90.87±0.47 | **91.47**±0.36 | 90.21±0.58 | 93.29±0.59 | **94.44**±0.53 | 92.62±0.67 |

*Table 7.* Performance comparison of different graph visualizers.

| | *Cora* | | | *Citeseer* | | |
| | Graphviz | Matplotlib | Igraph | Graphviz | Matplotlib | Igraph |
|---|---|---|---|---|---|---|
| $GVN_{NCNC}$ | 90.70±0.56 | 90.56±0.38 | **90.88**±0.66 | 94.12±0.58 | 93.96±0.48 | **94.28**±0.36 |
| $E\text{-}GVN_{NCNC}$ | **91.47**±0.36 | 91.42±0.24 | 91.31±0.58 | **94.44**±0.53 | 94.39±0.56 | 94.12±0.28 |

*Table 8.* Performance comparison of different vision encoders.

| | *Cora* | | | *Citeseer* | | |
| | ResNet50 | VGG | ViT | ResNet50 | VGG | ViT |
|---|---|---|---|---|---|---|
| $GVN_{NCNC}$ | **90.70**±0.56 | 89.99±1.61 | 90.69±0.44 | 94.12±0.58 | 93.93±1.35 | **94.29**±1.07 |
| $E\text{-}GVN_{NCNC}$ | **91.47**±0.36 | 89.92±1.01 | 91.24±0.66 | 94.44±0.53 | 93.96±0.85 | **94.52**±0.97 |

*Table 9.* Performance comparison of feature integration strategies.

| | *Cora* | | | *Citeseer* | | |
| | Attention | Concat | Weight | Attention | Concat | Weight |
|---|---|---|---|---|---|---|
| **$GVN_{NCNC}$** | **90.70**±0.56 | 85.65±6.25 | 89.99±1.64 | **94.12**±0.58 | 86.33±4.18 | 93.93±1.04 |
| **$E\text{-}GVN_{NCNC}$** | **91.47**±0.36 | 76.58±7.79 | 90.66±1.56 | **94.44**±0.53 | 88.25±5.54 | 93.88±1.21 |

*Table 10.* Performance comparison with varying VSF adaptivity.

| | *Cora* | | | *Citeseer* | | |
| | freeze | partial | full | freeze | partial | full |
|---|---|---|---|---|---|---|
| **$GVN_{NCNC}$** | 89.57±0.62 | 90.52±0.65 | **90.70**±0.56 | 91.55±0.79 | 93.75±0.75 | **94.12**±0.58 |
| **$E\text{-}GVN_{NCNC}$** | 89.66±0.54 | 91.47±0.36 | 91.53±0.55 | 92.72±0.48 | 94.44±0.53 | **94.52**±0.67 |

### 5.4. Scalability Analysis

Figure 5 compares the inference time and GPU memory for inferring one batch of samples from *Cora* (including pre-processing time). Among all the methods in comparison, GVN is the most time-consuming, followed by SEAL and NBFnet. These three methods also require considerably more memory than the others. This elevated resource consumption is due to the need for pre-processing and computation for each link, and the storage of intermediate variables with respect to links. Particularly, GVN requires graph visualization, which introduces extra pre-processing time. Therefore, similar to SEAL and NBFnet, GVN is not well-suited for large-scale graph computations. In contrast, the visualization of E-GVN is node-dependent, which can be reused for diverse links. After amortizing the pre-processing

visualization time across link queries, E-GVN still exhibits computational overhead similar to the base models GCN and NCNC. Therefore, when leveraging the lightweight base models (e.g., GCN or NCNC), E-GVN maintains efficiency for large-scale graphs.

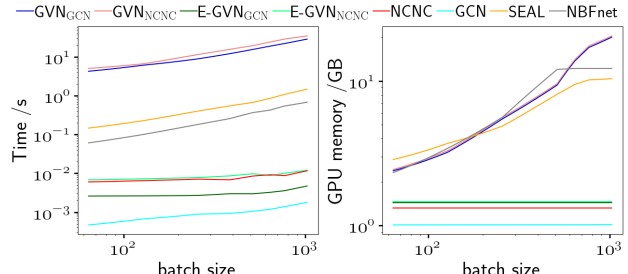

*Figure 5.* Inference time and the use of GPU memory of different methods on *Cora*.

### 5.5. Comparison with Positional Encodings

In addition to comparing with heuristic SFs, we further demonstrate the effectiveness of VSFs by comparing them against four representative encoding mechanisms for nodes: 1) 2-dimensional coordinates of nodes in the subgraph image, 2) Laplacian positional encoding, 3) Distances to other nodes, and 4) Node degree (centrality encoding). These positional encodings (PEs) are utilized as node features and processed by a 2-layer GCN followed by a 2-layer MLP for link prediction. The comparison results of Hits@100 score among VSFs and these encoding mechanisms are presented in Table 11. As observed, VSFs outperform the PEs across various datasets. Notably, directly encoding the 2-dimensional coordinates proves insufficient, highlighting the importance of employing a vision encoder to capture comprehensive structural information.

| | Cora | Citeseer | Pubmed |
|---|---|---|---|
| VSF | **71.32 ± 0.70** | **60.96 ± 0.68** | **48.27 ± 0.51** |
| 2-D Axis in Image | 38.57 ± 1.22 | 33.29 ± 1.80 | 25.34 ± 0.95 |
| Laplacian PE | 55.14 ± 0.84 | 52.03 ± 1.17 | 46.80 ± 0.75 |
| Distance | 42.03 ± 0.82 | 56.65 ± 0.43 | 44.21 ± 0.36 |
| Degree/Centrality | 42.80 ± 1.52 | 44.15 ± 1.61 | 34.90 ± 1.23 |

*Table 11.* Link prediction effectiveness between VSFs and positional encodings (Hits@100).

## 6. Conclusion

We propose GVN and E-GVN, novel frameworks that first reveal the vision potentials to enhance the link prediction ability for MPNNs. We provide key practices and address unexplored research questions in the integration of vision and MPNN. Comprehensive experiments demonstrate that the proposed methods significantly improve link prediction and are fully compatible with existing methods. GVN exhibits a novel direction for link prediction, with the potential to extend to more graph learning tasks.

## Impact Statement

This work underscores the often-overlooked yet effective role of visual perception in graph learning. By drawing attention to this aspect, it is poised to generate discussion and interest within the community. Additionally, it may inspire the development of further research that combines graph neural networks with visual insights, leveraging their synergy across a broader range of applications. For example, this approach could be extended to other graph tasks such as node classification, as well as applied to various types of graphs such as heterogeneous graphs and temporal graphs. We leave these for future works.

## Acknowledgement

This work was supported by National Key Research and Development Program of China (No. 2022ZD0160300), NSFC grant (No. 62136005), and the Research Grants Council of the Hong Kong Special Administrative Region (Grants 16200021, 16202523, C7004-22G-1). We also acknowledge the authors of (Chen & Wei, 2025; Lu et al.; 2024; 2025).

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

# A. Supplementary Algorithms

This section provides the algorithms for GVN and E-GVN.

## A.1. Algorithm for GVN with Selective Integration Strategies

---
**Algorithm 2** The GVN Framework.

---
**Require:** Graph $G = (V, E)$ with node attributes $\mathbf{X}_G$, queried links $\mathcal{Q} = \{(u, v)\}$.
**Ensure:** Link existence probabilities $\{p(u, v)|(u, v) \in \mathcal{Q}\}$
1:   $\mathbf{Y}_G \leftarrow \text{MPNN}(G, \mathbf{X}_G)$
2: **for** each $(u, v) \in \mathcal{Q}$ **do**
3:     **Step 1**: Scope-decoupled Subgraph Visualization
4:        $S_{uv}^k \leftarrow$ Extract $k$-hop subgraph around $(u, v)$
5:        $I_{uv}^k \leftarrow \text{GV}(S_{uv}^k)$
6:     **Step 2**: Adaptive Extraction of VSFs
7:        $\mathbf{v}_{uv} \leftarrow \text{VE}_\psi(I_{uv}^k)$
8:     **Step 3**: Compatible Features Integration
9:        $\mathbf{y}_u, \mathbf{y}_v \leftarrow \mathbf{Y}_G[u], \mathbf{Y}_G[v]$
10:    **if use** Attention-based Integration **then**
11:        $\tilde{\mathbf{v}}_{uv} \leftarrow \text{Linear}(\mathbf{v}_{uv})$
12:        $\tilde{\mathbf{y}}_u, \tilde{\mathbf{y}}_v \leftarrow \text{CA}(\mathbf{y}_u, \tilde{\mathbf{v}}_{uv}), \text{CA}(\mathbf{y}_v, \tilde{\mathbf{v}}_{uv})$
13:        $p(u, v) \leftarrow R(\tilde{\mathbf{y}}_u, \tilde{\mathbf{y}}_v)$
14:    **else if use** Concatenated Integration **then**
15:        $\tilde{\mathbf{y}}_u, \tilde{\mathbf{y}}_v \leftarrow \mathbf{y}_u \parallel \mathbf{v}_{uv}, \mathbf{y}_v \parallel \mathbf{v}_{uv}$
16:        $p(u, v) \leftarrow R(\tilde{\mathbf{y}}_u, \tilde{\mathbf{y}}_v)$
17:    **else if use** Weighted Integration **then**
18:        $p_{\text{vision}}(u, v) \leftarrow \text{VDecoder}(\mathbf{v}_{uv})$
19:        $p_{\text{MPNN}}(u, v) \leftarrow R(\mathbf{y}_u, \mathbf{y}_v)$
20:        $p(u, v) \leftarrow \delta \cdot p_{\text{vision}}(u, v) + (1 - \delta) \cdot p_{\text{MPNN}}(u, v)$
21:    **end if**
22:    Output $p(u, v)$
23: **end for**

---

## A.2. Algorithm for Efficient GVN (E-GVN)

---
**Algorithm 3** The Efficient GVN (E-GVN) Framework.

---
**Require:** Graph $G = (V, E)$ with node attributes $\mathbf{X}_G$, queried links $\mathcal{Q} = \{(u, v)\}$.
**Ensure:** Link existence probabilities $\{p(u, v)|(u, v) \in \mathcal{Q}\}$
1: **for** each node $v \in V$ **do**
2:     **Step 1**: Scope-decoupled Subgraph Visualization
3:        $S_v^k \leftarrow$ Extract $k$-hop subgraph around $(v)$
4:        $I_v^k \leftarrow \text{GV}(S_v^k)$
5:     **Step 2**: Adaptive Extraction of VSFs
6:        $\mathbf{v}_v \leftarrow \text{VE}(I_v^k)$
7:        $\tilde{\mathbf{v}}_v \leftarrow \text{Adaptor}_\phi(\mathbf{v}_v)$
8:     **Step 3**: Compatible Features Integration
9:    **if use** Attention-based Integration **then**
10:        $\tilde{\mathbf{x}}_v \leftarrow \text{CA}(\mathbf{x}_v, \tilde{\mathbf{v}}_v)$
11:    **else if use** Concatenated Integration **then**
12:        $\tilde{\mathbf{x}}_v \leftarrow \mathbf{x}_v \| \tilde{\mathbf{v}}_v$
13:    **else if use** Weighted Integration **then**
14:        $\tilde{\mathbf{x}}_v = \delta\text{Linear}_{\phi_1}(\mathbf{x}_v) + (1 - \delta)\text{Linear}_{\phi_2}(\tilde{\mathbf{v}}_v)$
15:    **end if**
16: **end for**
17: $\tilde{\mathbf{X}}_G \leftarrow \{\tilde{\mathbf{x}_v}\}$
18: $\tilde{\mathbf{Y}}_G \leftarrow \text{MPNN}(G, \tilde{\mathbf{X}}_G)$
19: **for** each $(u, v) \in \mathcal{Q}$ **do**
20:    $\tilde{\mathbf{y}}_u, \tilde{\mathbf{y}}_v \leftarrow \tilde{\mathbf{Y}}_G[u], \tilde{\mathbf{Y}}_G[v]$
21:    $p(u, v) \leftarrow R(\tilde{\mathbf{y}}_u, \tilde{\mathbf{y}}_v)$
22:    Output $p(u, v)$
23: **end for**

---

# B. Time Complexity Analysis

Let $n$ be the number of nodes, $d$ be the maximum node degree, $F$ be the node feature dimension, $F'$ be the dimension of node representation produced by MPNN, $S$ be the dimension of visual structural features (VSFs), and $l$ be the number of target links.

## B.1. Time Complexity Analysis for GVN

The time complexity of GVN is determined by the following components:

1. **Complexity of the base MPNN model**, which includes the MPNN and its associated read-out function. For example, the complexity of GCN is $O(ndF + nF^2) + O(lF^2)$. For the NCNC model (Wang et al., 2024) that incorporates common-neighbor SFs, the complexity is $O(ndF + nF^2) + O(ld^2F + ldF^2)$. We denote this part by $O(\text{MPNN})$.

2. **Complexity of generating visual images for the target links** is $O(l)$.

3. **Complexity of extracting visual structural features with Vision Encoder** is $l \cdot O(\text{VE})$, where $O(\text{VE})$ is the complexity of vision encoder.

4. **Complexity of feature integration**, which varies based on the integration strategy:
   - *Attention-based Integration*:
     - The linear projection that converts the $S$-dimensional VSFs $\mathbf{v}_{uv}$ to the $F'$-dimensional $\tilde{\mathbf{v}}_{uv}$ has a complexity of $O(lSF')$.
     - The cross-attention mechanism has a complexity of $O(lF'^2)$.
     
     Therefore, the total time complexity for attention-based integration is dominated by $O(lSF' + lF'^2)$.
   - *Concatenated Integration*:
     - Concatenating $\mathbf{v}_{uv}$ with $\mathbf{y}_u$ and $\mathbf{y}_v$ has a complexity of $O(l(F' + S))$, where $(F' + S)$ is the dimension after concatenation.
     
     Therefore, the total time complexity for Concatenated integration is dominated as $O(l(F' + S))$.
   - *Weighted Integration*:
     - Using VExpert (an MLP) to process $\mathbf{v}_{uv}$ has a complexity of $O(lS^2)$.
     - The weighted integration step has a complexity of $O(l)$.
     
     Therefore, the total time complexity for Weighted integration is dominated as $O(lS^2)$.

Thus, the total time complexity of GVN for each feature integration method is as follows:

- **Attention-based Integration**:
$$O(\text{MPNN}) + l \cdot O(\text{VE}) + O(l(SF' + F'^2)).$$

- **Concatenated Integration**:
$$O(\text{MPNN}) + l \cdot O(\text{VE}) + O(l(F' + S)).$$

- **Weighted Integration**:
$$O(\text{MPNN}) + l \cdot O(\text{VE}) + O(lS^2).$$

## B.2. Time Complexity Analysis for E-GVN

The time complexity components for E-GVN are as follows:

1. **Complexity of generating visual images for all nodes**: $O(n)$.

2. **Complexity of extracting visual structural features with Vision Encoder** is $n \cdot O(\text{VE})$.

3. **Complexity of processing VSFs with the Adaptor**: $O(nS^2)$.

4. **Complexity of integrating VSFs into node attributes before MPNN**:

   - *Attention-based Integration*:
     - The attention mechanism now operates on $n$ nodes instead of $l$ links, and operates on $F$ dimension node attributes instead of $F'$ dimension MPNN-produced node representations, resulting in a complexity of $O(nSF + nF^2)$.

     Therefore, the total time complexity for attention-based integration in E-GVN is $O(nSF + nF^2)$.

   - *Concatenated Integration*:
     - Concatenating $\mathbf{v}_v$ with $\mathbf{x}_v$ for all nodes has a complexity of $O(n(F + S))$.

     Therefore, the total time complexity for Concatenated integration in E-GVN is $O(n(F + S))$.

   - *Weighted Integration*:
     - Using separate Linear Experts to process $\mathbf{x}_v$ and $\mathbf{v}_v$ and for all nodes has a complexity of $O(nF^2 + nS^2)$.
     - The weighted integration step has a complexity of $O(n)$.

     Therefore, the total time complexity for Weighted integration in E-GVN is $O(nF^2 + nS^2)$.

5. **Complexity of the base model**: Remains $O(\text{MPNN})$.

Thus, the total time complexity of E-GVN for each feature integration method is as follows:

- **Attention-based Integration**:

$$O(\text{MPNN}) + n \cdot O(\text{VE}) + O(n(S^2 + SF + F^2)).$$

- **Concatenated Integration**:

$$O(\text{MPNN}) + n \cdot O(\text{VE}) + O(n(S^2 + F + S)).$$

- **Weighted Integration**:

$$O(\text{MPNN}) + n \cdot O(\text{VE}) + O(n(S^2 + F^2)).$$

## C. Dataset Statistics

The statistics of the datasets are shown in Table 12. Among these datasets, *Collab*,*PPA*, *Collab*, *DDI* and *Citation2* belong to large-scale graphs and the link prediction on them are more challenging.

For the Planetoid datasets (*Cora*, *Citeseer*, and *Pubmed*), since the official data splits are not available, we adopt the common random splits of 70%/10%/20% for training/validation/testing. For the OGB benchmarks *Collab*, *PPA*, *DDI*, and *Citation2* (Hu et al., 2020), we utilize the official fixed splits.

*Table 12.* Statistics of datasets.

|  | *Cora* | *Citeseer* | *Pubmed* | *Collab* | *PPA* | *DDI* | *Citation2* |
|---|---|---|---|---|---|---|---|
| #Nodes | 2,708 | 3,327 | 18,717 | 235,868 | 576,289 | 4,267 | 2,927,963 |
| #Edges | 5,278 | 4,676 | 44,327 | 1,285,465 | 30,326,273 | 1,334,889 | 30,561,187 |
| data set splits | random | random | random | fixed | fixed | fixed | fixed |
| average degree | 3.9 | 2.74 | 4.5 | 5.45 | 52.62 | 312.84 | 10.44 |

## D. Link Prediction Experimental Setups

In link prediction, links play dual roles: serving as supervision and acting as message-passing paths. Following the standard practice in link prediction, training links fulfill both supervision labels and message-passing paths. In terms of supervision, the training, validation, and testing links are mutually exclusive. For message passing, we follow the common setting (Li et al., 2024) where the validation links in *ogbl-collab* additionally function as message-passing paths during test time.

For the baselines, we directly use the results reported in (Wang et al., 2024) since we adopt the same experimental setup.

## E. Substructure Counting Experimental Details

The experimental settings of substructure counting in Table 1 are detailed as follows.

**Datasets Generation** We follow the experimental setup in (Chen et al., 2020) to generate two synthetic datasets of random unattributed graphs. The first one is a set of 5000 Erdos-Renyi random graphs, where each graph contains 10 nodes and each edge exists with the probability $p = 0.3$. The second one is a set of 5000 random regular graphs, where the number of nodes $m$ for each graph and the node degree $d$ is uniformly sampled from $\{(m = 10, d = 6), (m = 15, d = 6), (m = 20, d = 5), (m = 30, d = 5)\}$. As the ground-truth labels, we calculate the counts of two types of substructures, including 'Triangle' and '3-Star', both are common patterns in graph topology. The dataset split ratio is set as 30%/20%/50% for training/validation/testing, respectively.

**Models** We compare GCN (Kipf & Welling, 2017), VSF+GCN, SAGE (Hamilton et al., 2017), and VSF+SAGE. To predict the substructure count, we append a 3-layer MLP decoder after GCN or SAGE. For VSF+ "'GCN and VSF+SAGE, we concatenate the VSFs with the node representations produced by the MPNN, which are then used as inputs to the MLP decoder. All models are trained for 100 epochs using the Adam optimizer (Kingma, 2014), with learning rates searched from $\{1, 0.1, 0.05, 0.01\}$. The depths of GCN and SAGE are searched from $\{2, 3, 4, 5\}$, and the hidden dimensions are searched from $\{8, 32, 128, 256\}$. We select the best model based on the lowest MSE loss on the validation set to generate the results.

## F. Graph Visualizer

In our implementation, the default graph visualization processes for both GVN and E-GVN are developed using Graphviz. Graphviz (Gansner & North, 2000) is a powerful tool for creating visual representations of abstract graphs and networks. It allows for the customization of styles (e.g., shapes, colors, labels) of nodes and edges to tailor the visualization to specific requirements. By default, we use brown color to emphasize the centered link or node, white color for the background and to fill other nodes, and a rectangular shape for node outlines. We use the scalable force-directed placement algorithm (sfdp) as the default layout computation algorithm. All of these configurations are specified and implemented with Graphviz.

We also implement two other optional graph visualizers using Matplotlib (Tosi, 2009) and Igraph (Gabor Csardi, 2006). Matplotlib provides a versatile plotting library that can be used for basic graph visualization, offering customization of node and edge properties and integration with other data visualizations. Igraph, on the other hand, is specifically designed for network analysis and visualization, featuring a wide range of layout algorithms and the ability to handle large graphs efficiently.

These graph visualizers produce images with distinct stylistic characteristics, as illustrated in Figure 6. Specifically, Graphviz generates images that are typically more structured and clean, making them suitable for detailed analysis and professional presentations. Matplotlib is known for its flexibility, allowing for creative and highly customized images, which are ideal for exploratory data analysis. Igraph, on the other hand, focuses on revealing underlying patterns in complex networks, particularly excelling in handling large datasets with layout algorithms that effectively display the overall structure of the network.

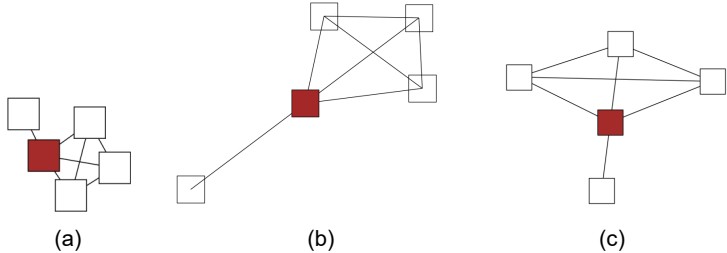

|       (a)       |       (b)       |       (c)       |

*Figure 6.* Node-centered examples of the visualized images by various visualizers (a) Graphviz (b) Matplotlib (c) Igraph.

## G. Style Consistency Ablation Experimental Details

In this section, we provide a detailed description of the experimental setup used in Table 5. This experiment demonstrates the negative impact of inconsistent styles on visualized subgraph images. Below, we outline the implementation details of

both consistent and inconsistent style settings.

The consistent style setting is achieved by using the default image style configuration as described in Appendix F. This involves maintaining uniformity in visual elements across all subgraph images, ensuring that each image adheres to the same stylistic parameters.

For the inconsistent style setting, configurations are randomly sampled from a predetermined range of options. This includes variations in node colors, shapes, and graph visualizers. The specific details are as follows:

- **Node Colors:** Colors are randomly selected from the set {White, Black, Brown, Yellow, Red, Green}. It is important to note that the centered link/node is colored distinctly from the other nodes to maintain focus.

- **Node Shapes:** Shapes are randomly chosen from the set {ellipse, box, circle, pentagon}.

- **Graph Visualizers:** The visualizer is randomly selected from the set {Graphviz, Matplotlib, Igraph}.

By employing these variations, the inconsistent style setting introduces heterogeneity in the visual representation of subgraphs, allowing us to assess the impact of style inconsistency on the overall analysis.

## H. Impact Study for Image Styles

### H.1. Impacts of Node Colors

In this section, we explore the effects of different color choices for node representations in graph images.

We first alter the colors of central nodes while keeping surrounding nodes white and evaluate performance on the Cora and Citeseer datasets (Hits@100). The results are summarized in Table 13.

*Table 13.* Performance (Hits@100) with Different Central Node Colors

| Center Node | GVN (Cora) | E-GVN (Cora) | GVN (Citeseer) | E-GVN (Citeseer) |
|---|---|---|---|---|
| Black | **90.72±0.52** | 91.43±0.31 | **94.12±0.58** | **94.46±0.52** |
| Brown | 90.70±0.56 | **91.47±0.36** | **94.12±0.58** | 94.44±0.53 |
| Dark Blue | 90.71±0.48 | 91.45±0.44 | 94.09±0.45 | 94.39±0.47 |
| Red | 90.66±0.50 | 91.40±0.40 | 94.00±0.50 | 94.30±0.50 |
| Green | 90.60±0.55 | 91.35±0.45 | 93.95±0.55 | 94.25±0.55 |
| Yellow | 90.68±0.57 | 91.42±0.38 | 94.05±0.57 | 94.40±0.54 |
| White | 89.35±0.72 | 89.90±0.65 | 93.20±0.72 | 93.55±0.75 |

From the above results, we have several findings:

**Findings 1** Only slight differences in performance when the model could distinguish central nodes from surrounding nodes. However, the model showed a preference for darker colors.

**Findings 2** When central nodes became white (indistinguishable from others), there was a noticeable performance degradation. This highlights the significance of labeling the identification of center nodes.

To further illustrate Findings 2, we assign colors to the nodes surrounding the central nodes. The results are presented in Table 14.

*Table 14.* Performance (Hits@100) with Different Surrounding Node Colors

| Center Node | Surrounding Node | GVN (Cora) | E-GVN (Cora) | GVN (Citeseer) | E-GVN (Citeseer) |
|---|---|---|---|---|---|
| Black | Black (same color) | 89.00±0.60 | 89.50±0.55 | 93.00±0.65 | 93.40±0.60 |
| White | White (same color) | 89.35±0.72 | 89.90±0.65 | 93.20±0.72 | 93.55±0.75 |
| Black | Brown (near color) | 90.20±0.50 | 90.70±0.45 | 93.80±0.55 | 94.10±0.50 |
| Black | White (opposite color) | **90.72±0.52** | **91.43±0.31** | **94.12±0.58** | **94.46±0.52** |

These results further reflect the preference of the model for more pronounced color differences between central and surrounding nodes, as indicated in Findings 2. The performance is lower when colors are the same or similar, and higher when there is a clear distinction.

## H.2. Impacts of Node Shapes

In this section, we investigate the impact of different node shapes on model performance. We experimented with three different shapes: Box, Circle, and Ellipse.

*Table 15.* Performance (Hits@100) with Different Node Shapes

| Center Node | GVN (Cora) | E-GVN (Cora) | GVN (Citeseer) | E-GVN (Citeseer) |
|---|---|---|---|---|
| Box | 90.70±0.56 | **91.47±0.36** | 94.12±0.58 | 94.44±0.53 |
| Circle | 90.65±0.52 | 91.40±0.38 | **94.15±0.43** | 94.42±0.50 |
| Ellipse | **90.72±0.46** | 91.45±0.44 | 94.10±0.57 | **94.46±0.52** |

According to Table 15, we find there is no obvious preference for a particular node shape.

## H.3. Impacts of Node Labeling Strategies

### H.3.1. SENSITIVE STUDY OF NODE LABELING STRATEGIES

*Table 16.* HR@100 performance with different node labeling strategies.

| | Cora | | | Citeseer | | |
|---|---|---|---|---|---|---|
| | No-label | Re-label | Unique | No-label | Re-label | Unique |
| $\text{GVN}_{NCNC}$ | **$90.70_{\pm 0.56}$** | $89.86_{\pm 0.44}$ | $89.67_{\pm 0.62}$ | **$94.12_{\pm 0.58}$** | $94.01_{\pm 0.43}$ | $94.08_{\pm 0.99}$ |
| $\text{E-GVN}_{NCNC}$ | **$91.47_{\pm 0.36}$** | $89.73_{\pm 0.24}$ | $89.75_{\pm 0.83}$ | **$94.44_{\pm 0.53}$** | $93.85_{\pm 0.65}$ | $94.02_{\pm 0.91}$ |

In this experiment, we study different ways to label the nodes in the image: (i) "No-label", which shows the nodes without any labels; (ii) "Re-label", which maps all the nodes in the current subgraph to new labels starting from zero; (iii) "Unique", which labels the nodes with unique global indices. Example images for these visualization strategies are shown in Appendix H.3.2. Table 16 shows the HR@100 performance of $\text{GVN}_{NCNC}$ and $\text{E-GVN}_{NCNC}$ with these different labeling strategies on *Cora* and *Citeseer*. As can be seen, "no-label" performs best, indicating that purely using the structural information is preferred.

### H.3.2. ILLUSTRATIONS FOR DIFFERENT LABELING STRATEGIES

Here, we present image examples for graph visualization used in both GVN and E-GVN with the three different labeling strategies.

Figures 7-9 show an example of link-centered subgraph visualization in GVN with various labeling strategies, where the target link is $(1, 158)$. This indicates the objective is to predict the existence of a link between node 1 and node 158. Similarly, Figures 10-12 present node-centered subgraph visualization images with various labeling strategies, where the colored node is the center node.

In Figures 7 and 10, the "No-label" labeling strategy is applied. In this strategy, node labels are omitted, enabling the model to focus purely on the intrinsic graph topological structural information, which is beneficial for generalizability across different datasets or settings.

Figures 8 and 11 adopt the "Re-label" labeling strategy, where the nodes within the subgraph are reassigned labels starting from 0. This local relabeling introduces some OCR noise, compelling the model to be more robust.

Finally, in Figures 9 and 12, which apply the "Unique" labeling strategy, nodes are labeled with their original IDs from the dataset. This might leverage the OCR capability to match nodes across various subgraphs due to the unique identifiers, which is beneficial for identifying node correspondences. However, this method may hamper generalizability and expose the model to the long-tail problem, where the model's performance degrades for nodes that appear infrequently in the data.

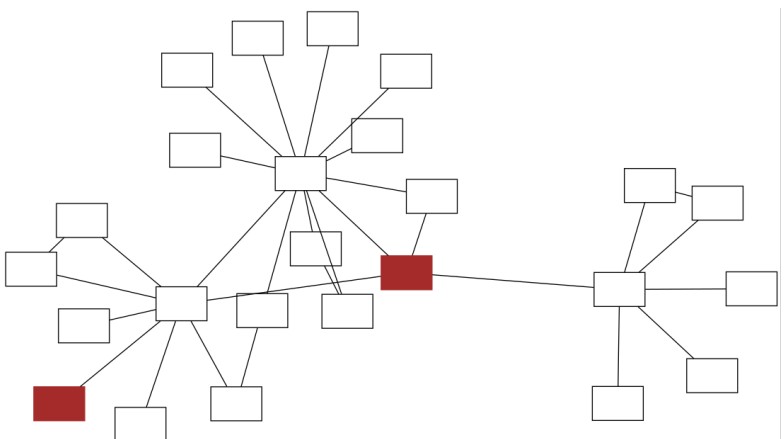

*Figure 7.* Link-centered subgraph visualization with "No-label" labeling scheme.

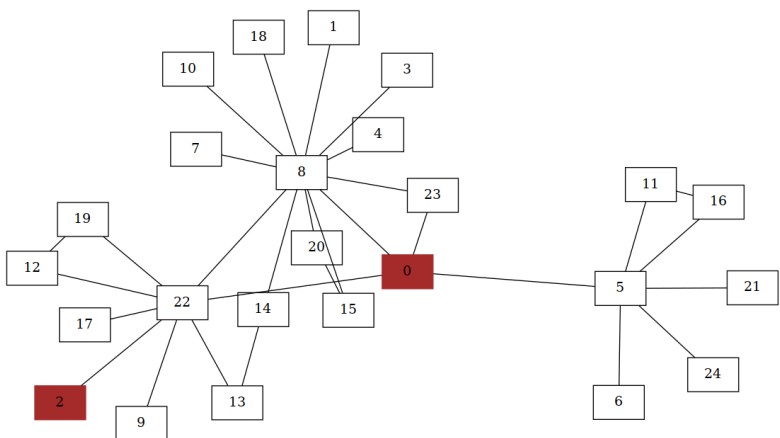

*Figure 8.* Link-centered subgraph visualization with "Re-label" labeling scheme.

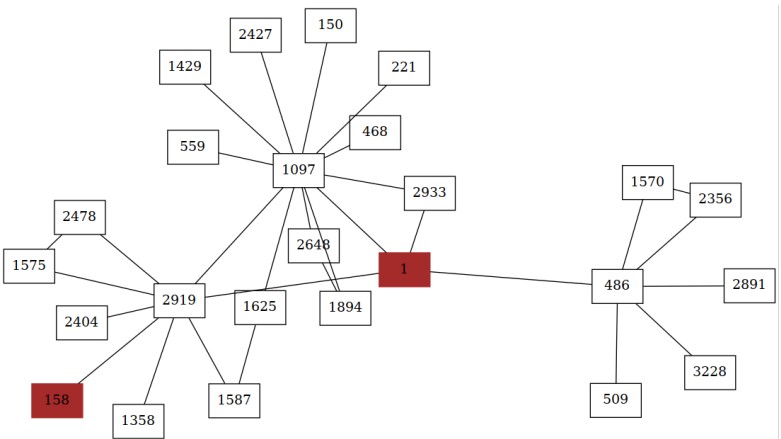

*Figure 9.* Link-centered subgraph visualization with "Unique" labeling scheme.

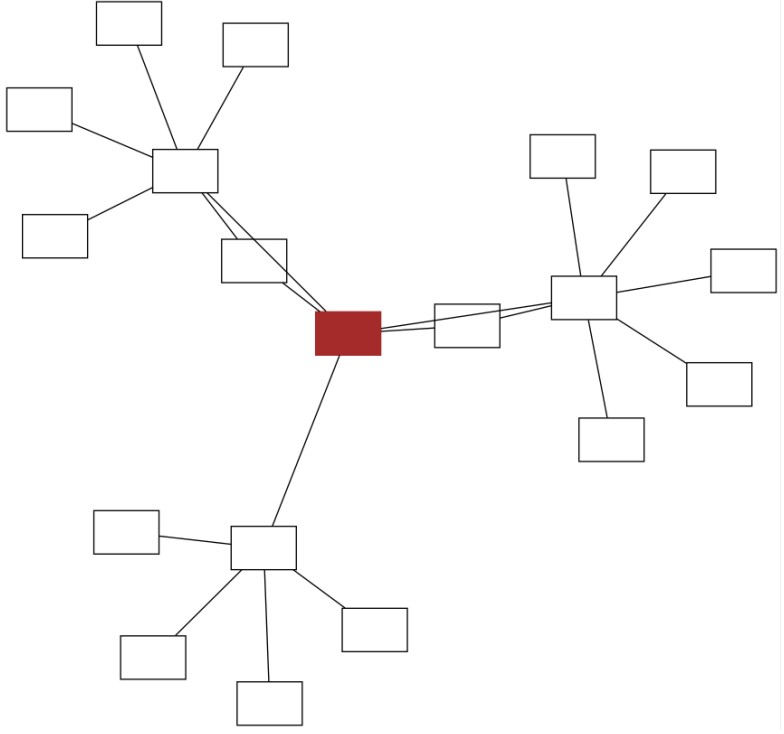

*Figure 10.* Node-centered subgraph visualization with "No-label" labeling scheme.

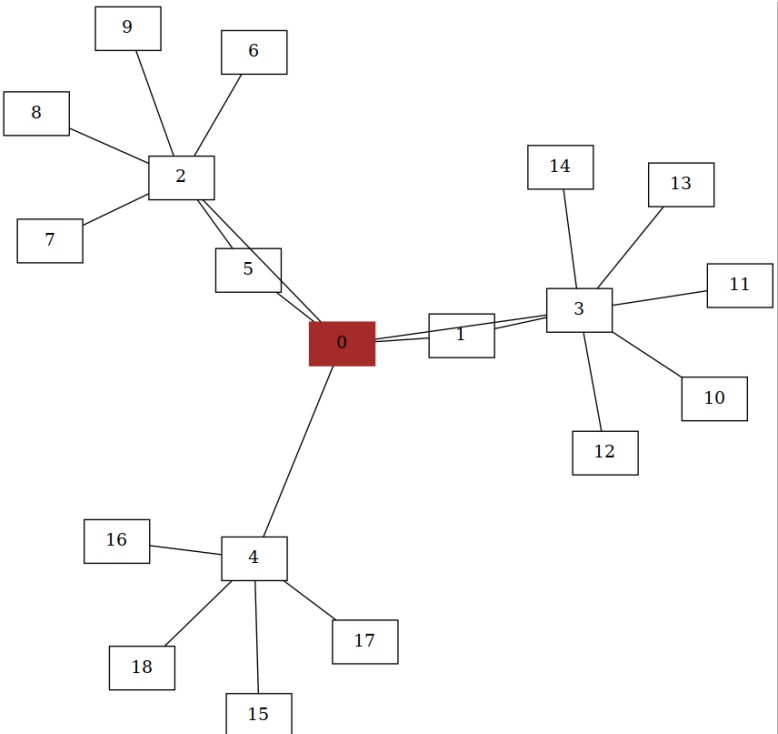

*Figure 11.* Node-centered subgraph visualization with "Re-label" labeling scheme.

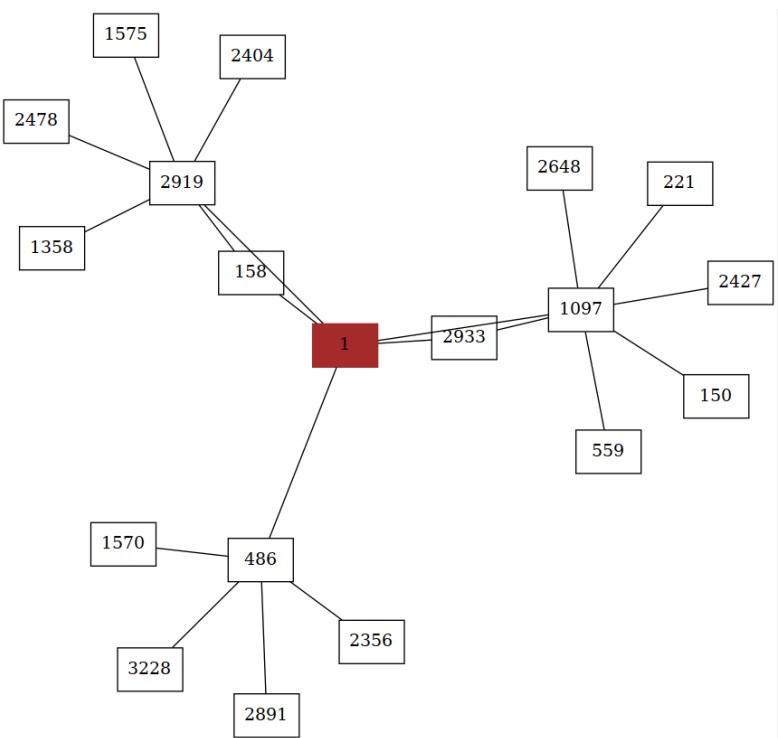

*Figure 12.* Node-centered subgraph visualization with "Unique" labeling scheme.

