# OpenReview forum: "Open Your Eyes: Vision Enhances Message Passing Neural Networks in Link Prediction"
_ICML.cc/2025/Conference — ICML 2025 poster_

### Official Review · Reviewer_cLaP · 2025-03-11

**Overall Recommendation:** 3

**Summary:**

The paper introduces a new GNN framework for link prediction tasks that can be used to extend existing architectures. The main idea is that the GNN can access image embeddings of visualizations of the (extended) neighborhoods. These are meant to enrich the representations with more context on where nodes are positioned in the graph and ultimately improve predictive performance. The paper claims that using this technique, state-of-the-art results can be achieved on common link prediction benchmarks.

**Claims And Evidence:**

One core claim the paper makes is that using image representations and embeddings from vision models (and the resulting “vision awareness”) improves performance. However, the paper never properly compares to other encoding techniques, like assigning the position of nodes in the visualization directly as features to nodes. The paper also fails to set the vision embeddings into context with other positional encodings commonly used for GNNs. While the performance seems to be good with the image embeddings, it’s not clear where these gains are coming from and whether one really needs to use vision models instead of some more simple and straight-forward encoding techniques.

**Essential References Not Discussed:**

In my opinion, this is one of the main weaknesses of the paper. The paper doesn’t mention any recent positional embeddings (for example, the very widely used Laplacian-based features) or other embeddings that were popularized together with graph transformers. Multimodal GNNs, such as those using language embeddings, could also be mentioned. Furthermore, and more importantly, the paper does not discuss graph visualization techniques and how they work. As this is at the core of this paper, I think it would be hugely important to explain what the employed visualization techniques try to optimize. This is its own research area and commonly referred to as “graph drawing”. The paper should mention some algorithms, at least the current state-of-the-art (which also includes some GNN models), and especially those that were used in the paper. The paper could also consider different optimization criteria like stress, edge crossings, angles, overlap between nodes, … and explain which ones were used for the visualizations. In the ablation study for different visualization techniques, the paper refers to them as “graphviz, matplotlib, and igraph”, which really doesn’t sufficiently describe what method was used, as graphviz alone has at least 7 very different algorithms to draw graphs. matplotlib and igraph are not even graph drawing frameworks, so it’s questionable what was used here. The Appendix claims that fdp was used for graphviz, is there any reason for this specific choice? Were the other methods compared to this?

**Experimental Designs Or Analyses:**

* The scalability analysis is generally a welcome addition, but the tests are only done on the smallest graph used in the benchmarking. Moreover, the scaling behavior is only analyzed by increasing the batch size. I’m really wondering: Is there any reason to believe that the methods do not scale linearly with increasing batch size? If the paper wants to analyze the scaling behavior to larger graphs, then one should increase the graph and not the batch size. This could (for example) be facilitated by sampling random graphs of increasing sizes and running the methods on them (for the scalability analysis it doesn’t really matter that much whether there is meaningful ground-truth data to learn).

* A comparison to other (more standard) encoding mechanisms for the node positions would be welcome, as well as a comparison to other standard positional encodings used for example for graph transformers.

**Methods And Evaluation Criteria:**

The paper proposes an MPNN architecture extension that could be used for any graph learning task. The paper does not convincingly explain why it chooses link prediction as the only task for evaluation. However, the datasets that are used in the evaluation are common and make sense.

**Other Comments Or Suggestions:**

I don't have more suggestions than the ones already mentioned.

**Other Strengths And Weaknesses:**

* GVs don’t usually yield unique and permutation equivariant representations

* It is not clear whether the visualization as an image brings any advantage over encoding the node positions (in the visualization) as features

**Questions For Authors:**

* Why does the paper only consider link prediction tasks?
* How do the vision embeddings perform in comparison to other positional encodings (like Laplacian-based ones, distance-based ones, and so on)?

## update after rebuttal
I appreciate the authors' responses and want to maintain my positive rating. At the same time, I still feel that limiting the evaluation to link prediction significantly weakens the paper, which is why I don't want to go higher.

**Relation To Broader Scientific Literature:**

The paper is grounded in the broader work on link prediction and extends existing methods to obtain better results.

**Theoretical Claims:**

The only theoretical claims are based around the runtime analysis and expressiveness of the model. Here, the paper does not mention the running time of O(VE), which, while dependent on the used visualizer, seems to be one of the major contributors and could be exemplified at least for the visualization method that is used in the final testing. Regarding the expressiveness (Remark 4.1): This result is not really surprising, as the method also completely breaks permutation equivariance. This fact is not really highlighted as a downside of the method, but would be important in this context.

---

> ### Author Rebuttal · Authors · 2025-03-31
>
> Thanks for your insightful reviews. **Please note that all the new tables and figures mentioned here are put in https://anonymous.4open.science/r/GVN-CLap/README.md**.
> > compares assigning the visualized node positions as node features.
> > compares more standard node positional encoding (PE) mechanisms like Laplacian-based and distance-based ones.
>
> As you adviced, we compared VSFs for nodes with 4 representative encoding mechanisms: 1) 2-d axis position of nodes in subgraph image, 2) Laplacian PE, 3) Distances to other nodes, and 4) Node degree (centrality encoding). These PEs were used as node features and decoded by 2-layer GCN+2-lyaer MLP for link prediction. The results in Table A show that **VSFs outperform those PEs across datasets**. Notably, directly encoding the 2-d axis is not effective enough, underscoring **the importance of using a vision encoder for comprehensive structural information**.
> > Why does the paper only consider link prediction tasks?
>
> As the title indicates, this work focuses on link prediction within the context of MPNN because:
> 1. Link prediction is one of the cornerstone tasks in graph learning, with 1) long history, 2) significant applications, and 3) well-established experimental settings, making it vital and representative.
> 2. As a first exploratory work for incorporating vision to MPNN, we faced many factors to explore. For example, we have revealed the effects on style consistency, node coloring, node labeling, image scopes, feature integration strategies, node shapes, visualizer, and encoder, etc. Efforts on those findings cost 8 months. Therefore, multiple efforts on other tasks are impractical for this starting work given limited resource.
>
> Moreover, we provide results in Table B, which demonstrate that **the benefits of VSF are promising for the node classification task**. We will study such extensions in future works.
> > Is there any reason to believe that the methods do not scale linearly with increasing batch size?
>
> While logarithmic axis labels are utilized in Fig. 5, the time actually scales linearly to the batch size. A version of Fig 5 with linear axis labels is provided in Fig. A.
> >  O(VE) should be exemplified.
> >  Welcome to analyzing the scalability on larger graphs by increasing the graph but not the batch size.
>
> As you advised, we here supplement another scalability analysis on randomly generated Erdos-Renyi graphs (edge possibility=0.2) with increasing numbers of nodes from 100 to 5000. The total training times for training 200 epochs of GCN/GVN/E-GVN/SEAL are provided in Table C, and the contributions of O(VE) are explicitly listed in parentheses. As shown, the O(VE) time of GVN scale quadratically w.r.t. the node size, making it unavailable for large graphs. Instead, E-GVN can handle large graphs in time comparable to GCN and its O(VE) time scales nearly linearly.
> > Remark 4.1 is not really surprising, as the method also completely breaks permutation equivariance.
> > GVs don’t usually yield unique and permutation equivariant representations.
>
> We acknowledge that VSFs are not permutation-equivariant. However, our performances have surpassed many permutation-equivariant SF-MPNNs, demonstrating it is not fatal. We think that there are two main reasons: 1) the learnable VSFs can provide some specific information, e.g., important structure patterns/motifs (Remark 4.2). 2) Visualization also imports meaningful data augmentations on permutation (e.g., varying layouts), which forces the decoder model to become less sensitive to node order, potentially alleviating the drawbacks. We will add these discussions with revisions.
> > Should mention representative works about 1) general positional encoder, 2）GNNs with language embeddings, 3) graph drawing techniques and their optimization targets
>
> Incorporating these discussions would definitely provide a more comprehensive background. We will add the discussion with them in the revision.
> > Matplotlib and igraph are not graph drawing frameworks. which doesn’t sufficiently describe what method was used
>
> Matplotlib is used to render images, while networkx handles the structures. For Igraph, it provides a built-in graph visualization function. We will add details about them in the revision.
> > Is there any reason for selecting fdp for graphviz?
>
> We choose fdp since a preliminary experiment comparing E-GVN with different layout algorithms on Cora (Table D) shows dot and fdp are better. Then we select fdp but not dot with reasons:
> 1. The force-based layout algorithm in fdp highlights node clustering, which is important for link prediction.
> 2. The tree-based layout in dot often results in a flattened image where the length exceeds the width, leading to a waste of the canvas.
>
> Thanks again for your kind reviews. We sincerely hope we have addressed your concerns. If you have any question, please feel free to discuss with us.

---

### Official Review · Reviewer_w7VD · 2025-03-13

**Overall Recommendation:** 3

**Summary:**

This paper proposes a novel framework called Graph Vision Network (GVN) and its efficient variant (E-GVN) to enhance link prediction in graph neural networks by integrating visual perception. The authors argue that while message-passing graph neural networks (MPNNs) and structural features (SFs) are dominant in link prediction tasks, the potential of visual perception has been overlooked.

**Claims And Evidence:**

The claims made in the paper are well-supported by extensive empirical evidence:
1. **Enhancement through Visual Awareness**: The authors claim that incorporating visual perception enhances link prediction performance. This is supported by experimental results showing significant improvements over baseline MPNNs and SF-enhanced MPNNs across multiple datasets.
2. **Compatibility with Existing Methods**: The paper demonstrates that GVN and E-GVN can be seamlessly integrated with existing models like GCN and NCNC, achieving new state-of-the-art results. This supports the claim that visual features provide orthogonal enhancements.
3. **Scalability**: The efficiency of E-GVN is demonstrated through reduced computational complexity and memory usage compared to GVN, making it suitable for large-scale graphs.

**Essential References Not Discussed:**

None

**Experimental Designs Or Analyses:**

The experimental designs are sound and comprehensive:
1. **Datasets**: The use of both small-scale (Planetoid) and large-scale (OGB) datasets ensures the evaluation covers a wide range of graph sizes and complexities.
2. **Baselines**: The comparison with strong baselines (e.g., GCN, NCNC) and various integration strategies provides a thorough analysis of the proposed methods' effectiveness.
3. **Ablation Studies**: The paper includes ablation studies on visualization styles, scopes, and adaptivity of VSFs, providing insights into the design choices and their impacts.

**Methods And Evaluation Criteria:**

The proposed methods and evaluation criteria are appropriate for the problem:
1. **GVN Framework**: The method involves converting subgraphs into visual representations using graph visualization tools and extracting visual structural features (VSFs) using a vision encoder. The integration of VSFs with MPNNs is explored through multiple strategies (attention-based, concatenated, and weighted). This approach is logical and well-motivated.
2. **Evaluation Criteria**: The authors use standard metrics (e.g., hit-ratio, MRR) for link prediction and evaluate on diverse datasets (Planetoid, OGB benchmarks). This ensures the robustness of their claims across different graph types and scales.

**Other Comments Or Suggestions:**

I think the authors should focus on the efficient GVN in the paper architecture, since it can get better performances with NCNC and much more efficient.

**Other Strengths And Weaknesses:**

Strength: Good performance and efficiency
Weakness: Since the method is a plugin outside the MPNN, the novelty of GNN structure is limited but not fatal. I will still give a positive score.

**Questions For Authors:**

None.

**Relation To Broader Scientific Literature:**

None

**Theoretical Claims:**

Not applicable.

---

> ### Author Rebuttal · Authors · 2025-03-31
>
> Thank you for your insightful feedback.
>
> > I believe the authors should concentrate on the efficient GVN within the paper's architecture, as it can achieve better performance with NCNC and is significantly more efficient.
>
> We sincerely appreciate your suggestion. We will focus on highlighting the efficient GVN in the revision.
>
> Thanks again for recognizing our work.

---

### Official Review · Reviewer_qvLH · 2025-03-15

**Overall Recommendation:** 1

**Summary:**

This paper proposes using visual structural features (VSFs) as a replacement for heuristic-based structural features (SFs) in graph learning tasks. The key contribution is the introduction of vision-based enhancements, which are empirically shown to improve message-passing neural network (MPNN) performance for link prediction.

**Claims And Evidence:**

The paper investigates whether and how visual awareness of graph structures benefits MPNNs in link prediction. The authors provide empirical evidence demonstrating improvements over baseline models in Section 5. However, the justification for why VSFs work better remains unclear, particularly in relation to existing expressive power analysis on models with SFs.

**Essential References Not Discussed:**

Several relevant works are missing from the discussion in Sections 4.3 and 4.4:
• BUDDY [1] and Bloom-MPNN [2] explored different strategies for integrating SFs with MPNNs, but they are not cited in relevant sections.
• SUREL [3] introduced the idea of decomposing query-induced subgraphs into node-level subgraphs for efficiency, which directly relates to the node-centered visualization approach in E-GVN (Section 4.4).

[1] Chamberlain, Benjamin Paul, et al. “Graph neural networks for link prediction with subgraph sketching.” ICLR’23.
[2] Zhang, Tianyi, et al. “Learning Scalable Structural Representations for Link Prediction with Bloom Signatures.” WWW’24.
[3] Yin, Haoteng, et al. “Algorithm and system co-design for efficient subgraph-based graph representation learning.” VLDB’22.

**Experimental Designs Or Analyses:**

The scalability analysis is incomplete. While inference time and GPU memory usage are reported, key details such as the configuration of VSF and the actual preprocessing overhead of VSFs are missing. Figure 5 lacks clarity regarding which encoder and number of hops for VSF were used. Since the primary computational cost comes from VSFs, explicitly reporting their preprocessing runtime would provide a clearer picture of the method’s efficiency.

**Methods And Evaluation Criteria:**

The paper utilizes standard link prediction benchmarks for performance comparison, making its evaluation broadly relevant. However, more details on how hyperparameters and configurations were chosen would strengthen the evaluation.

**Other Comments Or Suggestions:**

N/A

**Other Strengths And Weaknesses:**

• Strengths: The idea of leveraging vision models for structural feature extraction is straightforward and empirically effective based on the reported results.
• Weaknesses: The paper lacks a clear justification linking the empirical improvements of VSFs to existing analyses of SF-based methods. Additionally, the increased computational complexity—due to large vision models such as ViTs—raises concerns about whether the performance gains justify the additional overhead.

**Questions For Authors:**

1. How does VSF handle hub nodes, such as influencers in social networks or highly cited papers? Could the limited scope of visual perception cause nodes with non-isomorphic subgraphs to be mapped to non-distinguishable VSF?
2. How well does VDecoder perform as a standalone model on benchmark datasets?
3. Considering VSF requires style consistency, wouldn’t node-centered subgraph visualization degenerate back to standard MPNNs for nodes with isomorphic subgraphs?
4. Could the authors provide a direct runtime comparison of preprocessing VSFs vs. classical SFs? Additionally, what are the vision model configurations and hyperparameters used in Table 3, Figure 5, and other results in Section 5?

**Relation To Broader Scientific Literature:**

This paper presents an interesting adaptation of visual features to enhance MPNN-based link prediction.

**Theoretical Claims:**

The theoretical justification for VSFs in Section 4.2 is grounded in prior work on subgraph-based methods (e.g., SEAL and labeling tricks). However, the expressive power of models incorporating VSFs remains unclear, especially given their black-box nature and potential randomness in mapping and encoding. This raises concerns about whether VSFs fundamentally improve representation power or simply introduce additional complexity without well-characterized benefits.

---

> ### Author Rebuttal · Authors · 2025-03-31
>
> Thanks for your insightful reviews. We here address your concerns point by point by merging and re-arranging relevant comments together:
> > The justification for why VSFs work better remains unclear, particularly in relation to existing expressive power analysis on models with SFs.
> > The expressive power of models incorporating VSFs remains unclear,  raises concerns about whether VSFs fundamentally improve representation power or simply introduce additional complexity without well-characterized benefits.
> > How well does VDecoder perform alone?
>
> First, we want to clarify that we **did not claim VSFs always have stronger expressive power than other SFs**, instead, our main claims for VSFs include
> 1. **When used alone, VSFs benefits MPNNs**. (Qualitatively demonstrated in Sec. 4.2, remarks 4.1-4.4).
> 2. **When used together with other SFs, VSFs still provide remarkable orthogonal improvements.** (Empirically demonstrated in Sec. 5).
>
> Second, the proposed VSFs show different characteristics from classic SFs: 1) Classic link prediction SFs are typically derived from a single type of heuristic (e.g., SFs in SEAL only encode path distance, and SFs in BUDDY only focus on high-order common neighbors). However, VSFs encode the whole subgraph from images, containing **massive and hybrid types** of structural biases (as demonstrated in Fig.4). 2) With training on specific data, VSFs **can vary** to align the data requirements (Remark 4.4). Due to such characteristics, it is not easy to theoretically analyze the properties of VSFs precisely, and instead we conducted empirical demonstrations (Sec. 4.2 and 5) to describe properties of VSFs.
>
> Lastly, **we provide the standalone performance of 2-hop VSFs from ResNet50 encoder in the following table, which is a direct comparison for the representation quality of different SFs on the link prediction task.** As illustrated, both individual link-based and node-based VSFs proposed in this paper outperform common neighbor SFs (i.e., CN, RA, and AA) and distance-based SFs (i.e., SPD) across various datasets, showing their superior structural representations beyond classic link prediction SFs.
>
> |SFs|Cora|Citeseer|Pubmed|
> |---|---|---|---|
> |CN|33.92±0.46|29.79±0.90|23.13±0.15|
> |AA|39.85±1.34|35.19±1.34|27.38±0.11|
> |RA|41.07±0.48|33.56±0.17|27.03±0.35|
> |SPD|29.97±0.76|41.37±0.81|40.21±0.59|
> |VSF(link)|**68.24±0.41**|**57.66±0.47**|**46.81±1.02**|
> |VSF(node)|65.98±0.72|55.51±0.91|45.24±0.44|
> > Implementation details and hyperparameter settings for Sec 5’s results are missing, which makes reproducing results difficult.
> > What are the vision model configurations and hyperparameters used in Table 3, Fig. 5 and other results.
> > Explicitly provide a direct runtime comparison of preprocessing VSFs vs. classical SFs.
>
> Thanks for your suggestion.
> 1. For hyperparameter configurations, please note that we have included **all the hyperparameters for the main results in ./scripts/ path of our submitted code repo. We also give the ranges and optimization approaches of the significant hyperparameters in Sec 5.1. Therefore, we think that it is easy to reproduce our results**. For the scalability analysis in Fig.5 and other experiments, the encoder is ResNet50 by default as mentioned in Sec. 4.2, and the default hop is 2.
> 2. Pre-processing time compared with SFs: We provide the time comparison on pre-processing VSFs and classic SFs on cora and ogbl-ddi testing in the following table. As shown, **E-GVN can process VSFs in comparable time (in seconds) as classic SFs. For the dense graph like ogbl-ddi, it is even more efficient** because the number of nodes are much smaller than that of links.
>
> |Dataset|VSF(GVN)|VSF(E-GVN)|CN|RA|AA|SPD|
> |---|---|---|---|---|---|---|
> |Cora|2.91e1|5.02e-2|**1.69e-2**|1.81e-2|1.89e-2|3.06e-2|
> |DDI|4.39e4|**6.20e0**|1.46e2|1.95e2|2.07e2|4.46e1|
> > How does VSF handle hub nodes?
>
> As exemplified in Fig 2, hub nodes are faithfully reflected in the image, revealing important cluster information.
> > Could the limited scope of visual perception cause nodes with non-isomorphic subgraphs to be mapped to non-distinguishable VSF?
>
> No. For any hop k, non-isomorphic subgraphs are visualized as different images, resulting in distinguishable VSFs.
> > Would node-centered visualization degenerate to standard MPNNs for nodes with isomorphic subgraphs?
>
> No. Though we eliminated node labels for structural clarity in visualization, we constructed the adjacent matrix with ascending order of node labels. This makes the visualization for isomorphic subgraphs different in layout if their relative label orderings are not permutation-equivariant.
> > Several relevant works are missing from the discussion in Sec 4.3 and 4.4.
>
> Thanks for your supplement. Incorporating these discussions can definitely make the discussion more comprehensive. We will add discussions with them in the revision.
>
> Sincerely wish we could address your concerns. For any further clarification, please feel free to let us know.

---

### Official Review · Reviewer_1KMA · 2025-03-19

**Overall Recommendation:** 3

**Summary:**

This paper proposed to incorprate vision information into MPNN to enhance link prediction. Specfically, it designed two framework, Graph Vision Network(GVN),along with a more efficient variant (E-GVN).

**Claims And Evidence:**

This paper analyzes the potential benefits to incorporate vision awareness in link prediciton and provides some empirical evidences.
This paper also demonstrated the effectiveness of the proposed GVN and E-GVN across several link prediction datasets, and empirically show the vision awarness can bring orthogonal improvements to SOTA methods.

**Essential References Not Discussed:**

NA

**Experimental Designs Or Analyses:**

The experimental designs are overall thorough and convincing.

**Methods And Evaluation Criteria:**

Yes, the empirical study can well support the effectiveness of the proposed method.

**Other Comments Or Suggestions:**

1. The discussion for RQ1 could be more convincing if some empirical study or theoretical analysis could be provided, instead of just intuition analysis.
2. More link prediction methods that considering the graph sub-structures could be included in the comparison, such as Link-MOE ("Mixture of Link Predictors on Graphs").

**Other Strengths And Weaknesses:**

Pros:
1. This paper proposed a novel idea to incorprate the vision modality into MPNN for link prediction, and demonstrate its rationality and effectiveness via empirical study.
2. Most of the intuitions are well supported and discussed via empirical study and preliminary analysis.
3. The experiments are carefully designed from different perspectives to support its design and motivations.

Cons:
1. The discussion for RQ1 could be more convincing if some empirical study or theoretical analysis could be provided, instead of just intuition analysis.
2. More link prediction methods that considering the graph sub-structures could be included in the comparison.

**Questions For Authors:**

NA

**Relation To Broader Scientific Literature:**

This paper proposed that incorporating the vision modality into MPNN can enhance link prediction, and support it via emipiral observations and experrimental studies.

**Theoretical Claims:**

NA

---

> ### Author Rebuttal · Authors · 2025-03-31
>
> Thanks for your insightful reviews.
> > 1. The discussion for RQ1 could be more convincing if some empirical study or theoretical analysis could be provided, instead of just intuition analysis.
>
> Thanks for your advice. For RQ1, you can find the following supports for our discussion.
> 1) For subgraph visualization, we have provided empirical ablation for **keeping style consistency** in Table 4 of Sec 5.3, where Appendix H provides more details. Appendices I.1 and I.2 provide the empirical study on color and shape influences on nodes, particularly demonstrating the significance of **highlighting the center nodes**. Appendix I.3 provides an ablation study on **node labeling strategies**, noting that eliminating node labels performs the best.
> 2) For decoupled vision scope, we exemplified the actual cases in Figure 2 to support our claim about the vision scope and provide a sensitivity study in Table 5 of Sec 5.3. The theoretical analysis of decoupling subgraph scope from MPNN has been proven by [r1], and we omitted the details due to the page limit. We will add them in the revision.
>
> [r1] Decoupling the depth and scope of graph neural networks. NeurIPS 2021
> > 2. More link prediction methods that consider the graph sub-structures could be included in the comparison, such as Link-MOE ("Mixture of Link Predictors on Graphs").
>
> As suggested, we have included Link-MOE in the comparison. Please note that Link-MOE integrates many MPNNs as experts to achieve good performance. Differently, as a plugin method outside MPNN, our proposed framework can be a supplement for enhancing **individual** MPNN to provide additional enhancements. Some results (i.e., Hits@50 for ogbl-collab, Hits@100 for ogbl-ppa) are shown in the following table, where we substitute the GCN and NCNC experts in Link-MOE as E-GVN(GCN) and E-GVN(NCNC). According to the results, we can see that incorporating VSFs can further enhance the performance of Link-MOE.
>
> ||ogbl-collab|ogbl-ppa|
> |---|---|---|
> |Link-MOE|71.32 ± 0.99|69.39 ± 0.61|
> |Link-MOE+VSFs|71.84±0.85|70.06±0.56|

---

### Decision · Program_Chairs · 2025-05-01

**Decision:**

Accept (poster)

**Comment:**

This paper introduces Graph Vision Network (GVN, link-centered visualization) and its efficient variant E-GVN (node-centered visualization), which integrate visual structural features into MPNNs to enhance link prediction. This is the first work to leverage graph visualizations as input to vision models (e.g., ResNet, ViT) for structural feature extraction and show improvement to state-of-the-art link prediction methods by achieving SOTA results on 7 LP benchmarks, demonstrating orthogonal improvements when combined with existing methods (e.g., GCN, NCNC). Some light concerns are raised regarding theoretical expressivity, comparison to recent PE methods, break of permutation equivariance, etc.

I think this work opens a new direction for multimodal GNNs, and its empirical gains warrant attention despite methodological gaps. I encourage the authors to add discussions on 1) positional encoding comparisons (Laplacian, distance-based), 2) permutation equivariance trade-offs and other limitations of the current strategy, and 3) make E-GVN the main focus due to its better efficiency and performance.